



**Convection-Aerosol Interactions in the United Arab Emirates: A Sensitivity Study**
Ricardo Fonseca[1], Diana Francis[1*], Michael Weston[1], Narendra Nelli[1], Sufian Farah[2],
Youssef Wehbe[2], Taha AlHosari[2], Oriol Teixido[3], Ruqaya Mohamed[3].
[1]Khalifa University of Science and Technology, P. O. Box 54224, Abu Dhabi, United Arab Emirates.
[2] National Center for Meteorology (NCM), P. O. Box 4815, Abu Dhabi, United Arab Emirates.
[3]Environment Agency – Abu Dhabi (EAD), P.O Box 45553, Abu Dhabi, United Arab Emirates.
*Correspondence to*: Diana Francis (diana.francis@ku.ac.ae)
**Abstract:**
The Weather Research and Forecasting (WRF) model is used to investigate convection-aerosol interactions
in the United Arab Emirates for a summertime convective event. Both an idealised and scaled versions of
a 7-year climatological aerosol distribution are considered. The convection on 14 August 2013 was
triggered by the low-level convergence of the circulation associated with the Arabian Heat Low (AHL) and
the daytime sea-breeze circulation. The cold pools associated with the convective events, as well as the
low-level wind convergence along the Intertropical Discontinuity (ITD) earlier in the day, explain the
dustier environment, with Aerosol Optical Depths (AODs) in excess of two.
Due to a colder surface and air temperature, the AHL is incorrectly represented in WRF, which leads to a
mismatch between the observed and modelled clouds and precipitation. Employing interior nudging in the
outermost grids of the three-nested simulation has a small but positive impact on the model predictions of
the innermost nest. This is because the higher temperatures from more accurate boundary conditions are
offset by colder temperatures from locally enhanced precipitation, the latter arising from a shift in the
position of the AHL. Numerical experiments revealed a high sensitivity to the aerosol properties. In
particular, replacing 20% of the rural aerosols by carbonaceous particles has an impact on the surface
radiative fluxes comparable to increasing the aerosol loading by a factor of 10, with a daily-averaged
reduction in the UAE-averaged net shortwave radiation flux of ~90 W m$^{-2}$ and an increase in the net



longwave radiation flux of $\sim 51\,\mathrm{W\,m^{-2}}$. However, in the former, WRF generates 20% more precipitation than
in the latter, due to a broader and weaker AHL.
The surface downward and upward shortwave and upward longwave radiation fluxes are found to scale
linearly with the aerosol loading, while the downward longwave radiation flux varies by less than $\pm\,12\,\mathrm{W}$
$\mathrm{m^{-2}}$ when the aerosol amount and/or properties are changed. An increase in the aerosol loading also leads
to drier conditions due to a shift in the position of the AHL and rainfall occurring in a drier region, with a
domain-wise decrease in the daily accumulated rainfall of 16% when the aerosol loading is increased by a
factor of 10. In addition, the onset of convection is also delayed.
**Keywords**:
Convection, Cold Pools, Aerosols, Numerical Modelling, Grid Nudging, United Arab Emirates.





## 1. Introduction

It has long been known that aerosols, defined as solid or liquid particles suspended in the atmosphere, both from natural and anthropogenic (human) sources, play an important role in the climate system (e.g. Ramanathan et al., 2001; Choobari et al., 2014; Boucher, 2015). Aerosols interact both with the radiation (direct and semi-direct effects; Satheesh and Moorthy, 2005) and cloud microphysics (indirect effects; Lohmann and Feichter, 2005). For simplicity, the former will be denoted as aerosol-radiation interactions (ARI) and the latter as aerosol-cloud interactions (ACI) throughout the text. Aerosols scatter and absorb solar (shortwave) and thermal (longwave) radiation, leading to a warming of the aerosol layer and a cooling of the surface below. As far as the ACI effects are concerned, an increase in aerosol loading leads to a larger number of smaller cloud droplets (first indirect or Twomey effect), which translates into a higher cloud albedo and optical depth, as smaller and more numerous particles reduce the "radiative windows to space". As a result, aerosols act to suppress precipitation, increasing the cloud lifetime and cloud height (second indirect or Albrecht effect). While pollution and smoke from industrial activities are the most common anthropogenic aerosols, dust is the most abundant natural aerosol on Earth. The Sahara Desert is the main source region of mineral dust, with contributions from other hyperarid regions such as the Arabian Desert in the Middle East (Francis et al., 2019b), the Gobi Desert in East Asia, and the Sonoran Desert in the United States (Tegen and Schepanski, 2009). Dust has been shown to have an important impact on the climate system, in particular on the atmosphere (e.g. Min et al., 2014; Liu et al., 2019; Francis et al., 2020), ocean (e.g. Evan et al., 2012) and cryosphere (e.g. Francis et al., 2018) dynamics.

The direct and indirect effects of dust aerosols on convection are discussed in Huang et al. (2019) for a Mesoscale Convective System (MCS) that developed over North Africa in July 2010 using the Weather Research and Forecasting (WRF; Skamarock et al., 2019) model. The authors conducted four simulations, switching on/off the dust-radiation and dust-cloud feedbacks. The ACI effects initially weaken the convective system, due to the slowdown of the conversion rate from cloud to rain and subsequent suppression of warm rain formation, but later strengthen it, as dust acts as condensation nuclei and increases



the amount of hydrometeors. In the end, there is a roughly 18% increase in precipitation with respect to the
simulation where the feedback is not activated. The ARI effects are found to have the largest influence on
the development of convection in dusty areas, leading to a stronger, albeit delayed, MCS. This is because
the heating of the dust layer during the day reduces convective instability, but the increase in downward
longwave radiation flux at the surface (e.g. Francis et al., 2020) will ultimately lead to higher values of
Convective Available Potential Energy (CAPE), and a roughly 14% increase in the accumulated
precipitation. In other words, it takes longer for the storm to develop, but when it does, it makes use of the
increased CAPE, which builds up over time, producing a stronger and longer-lived MCS. When ACI is
added, the MCS intensifies further, with the increase in total rainfall as high as 39% during the first
convective development cycle. This figure is larger than the sum of the precipitation increase when the ARI
and ACI effects are switched on separately, evidence of a non-linear interaction of the two effects. In these
simulations the dust was lifted from the surface in barren or sparsely vegetated areas, when the wind speed
exceeded the critical threshold of 6 m s$^{-1}$. Liu et al. (2020) used the WRF model with Chemistry (WRF-
Chem; Grell et al., 2005) to investigate the effects of biomass burning aerosols on radiation, clouds and
precipitation in the Amazon basin. The authors found that ACI effects prevail at lower emission rates and
low values of aerosol optical depth (AOD), while the ARI plays the largest role at high emission rates and
high AODs. Regarding the precipitation, the presence of biomass burning aerosols leads to lower
precipitation rates and frequency of occurrence, with the ACI feedback playing the largest role at low
aerosol loading, and the ARI effect being the dominant feedback at high aerosol loading. Menut et al. (2019)
tested the sensitivity of the WRF response to anthropogenic and mineral dust emissions over the Sahara for
July 2016. They concluded that dust played a larger role in terms of ACI effects, with a doubling of its
amount leading to a 0.5 K and 25 m decrease in the 2-meter temperature and the planetary boundary layer
(PBL) depth, respectively. The surface net shortwave and longwave radiation fluxes changed by up to 25
W m$^{-2}$, the former decreasing and the latter increasing. However, a drop in the anthropogenic emissions
along the African coast led to a northward shift of the monsoon precipitation, with increased near-surface
winds (and hence dust emissions) over the desert areas. In other words, aerosols can also induce a change





in the regional atmospheric circulation. When the model predictions are evaluated against observations,
some authors found that accounting for the ACI and ARI effects clearly improves the accuracy of the
forecasts, e.g. Thomas et al. (2021) for a rainfall event in India, while others reported a smaller impact, e.g.
Lompar et al. (2019) for a summertime convective event in Serbia. Adding the effects of aerosols also
improves the model representation of clouds, for both ice- and liquid-water related quantities (e.g. Su and
Fung, 2018; Glotfelty et al., 2019).

The United Arab Emirates (UAE) is a country located in the Middle East, bounded by the Arabian Gulf
to the north and west, the Sea of Oman to the northeast, and the Rub' Al Khali desert to the south. The
country is rather flat with an elevation of typically less than 300 m above mean sea level, except in the
northeastern side where the Al Hajar mountain range dominates the landscape, with the highest elevation
of around 2,000 m at Jabel Jais. The meager and irregular amounts of precipitation, which range from less
than 40 mm in the southern desert to over 120 mm over the mountains, mostly fall in the cold season from
November to March, in association with mid-latitude weather systems (Niranjan Kumar and Ouarda, 2014;
Wehbe et al. 2017, 2018). However, summertime convective events are present as well, and can lead to
rainfall accumulations of more than 100 mm and flash floods at isolated sites (Steinhoff et al., 2018; Branch
et al., 2020; Wehbe et al., 2020; Francis et al., 2021).

Convection in the warm season in the UAE tends to take place on the eastern half of the country, around
the Al Hajar mountains. As discussed in Schwitalla et al. (2020) and Branch et al. (2020), it is normally
triggered by the convergence of the low-level circulation associated with the Arabian Heat Low (AHL;
Fonseca et al., 2021), the sea-breeze circulation from the Arabian Gulf and Sea of Oman, and the upslope
flows on the mountains. The presence of a mid- to upper-level trough, originating from the mid-latitudes,
and associated unstable stratification also promotes the development of convective clouds (Francis et al.,
2021). An inspection of satellite data revealed an average of 55 of such events per summer season, peaking
in July, followed by August and June (Branch et al., 2020). The convection typically initiates at elevations



of 600 to 800 m on the oceanic side of the mountains around 13-15 local time (LT), propagating northwards,
southwards and westwards during the afternoon hours, in response to the background flow and local
topography. Although less frequent, convective events also take place in the flatter western half. Here, they
are commonly triggered by the low-level convergence of the AHL and sea-breeze circulations (Steinhoff et
al., 2018). The AHL is a shallow, warm-core, cyclonic system that develops in response to the strong
heating of the surface by the Sun in the Arabian Peninsula (Racz and Smith, 1990). Its strength is modulated
by the Indian Summer Monsoon (Steinhoff et al., 2018), sea surface temperatures (SSTs) in the Indian
Ocean (Yu et al., 2015) and in the equatorial Pacific (Fonseca et al., 2021). A stronger AHL, typically seen
during periods of enhanced convective activity over the Arabian Sea, as the increased descent and
subsidence over the Arabian Peninsula helps to intensity the heat low, modulates the inland penetration of
the marine boundary layer. The convergence line between the more moist marine air and the hotter and
drier desert air, which plays an important role in the triggering of dust (e.g. Dumka et al., 2019; Rashki et
al., 2019) and convective (e.g. Francis et al., 2020) storms, is labelled as the Intertropical Discontinuity
(ITD). Its position is therefore linked to the strength and spatial extent of the AHL, as explained in Fonseca
et al. (2021).

Being part of the Arabian Desert, aerosols are ubiquitous in the UAE. As discussed in Nelli et al. (2021),
the prevailing aerosol subtype is dust, with anthropogenic aerosols present mostly in the cold season,
advected by the background (northwesterly) winds. On seasonal time-scales, the AOD is higher in summer
and spring, typically in the range 0.3-0.6, with a secondary peak in February, likely associated with the
passage of mid-latitude baroclinic systems. During dust storms, on the other hand, the AOD can exceed the
climatologically averaged values by an order of magnitude: e.g. during the July 2018 event, the AOD
exceeded 3 with more than $20\times10^{15}$ g (20 Tg) of dust being lifted into the atmosphere (Francis et al., 2020).
On diurnal scales, the AOD values are slightly higher in the early morning when the nighttime low-level
jet mixes down to the surface, with the stronger near-surface winds lifting higher amounts of dust (Bou
Karam Francis et al., 2017). The aerosol variability in the UAE is also discussed in Kesti et al. (2021),





which analyses measurements collected by a Lidar deployed at Al Dhaid, a city roughly 80 km to the
northeast of Dubai, from February 2018 to February 2019. The authors concluded that the size of the
aerosols is more important than their chemistry (i.e. composition, which affects the hygroscopicity) for
aerosol particle activation, in line with the findings of Dusek et al. (2006).

In this work, the interaction between aerosols and convection in the UAE is investigated for a
summertime convective event that took place on a relatively dusty day. This is achieved through a set of
sensitivity experiments with the WRF model, using both idealised and climatological aerosol distributions,
the latter scaled so as to improve the agreement with in-situ measurements. Despite a persistent cold bias,
WRF has been found to perform well for summertime convective events in the UAE (e.g. Steinhoff et al.,
2018; Schwitalla et al., 2020; Francis et al., 2021). The two main objectives of this study are as follows: (i)
investigate the added value of incorporating aerosols and accounting for its direct and indirect effects on
the model-predicted convective activity, and (ii) explore the sensitivity of the WRF response to different
aerosol loadings and properties. The findings of this work will be very relevant to other arid/hyperarid
regions, in particular those adjacent to major aerosol sources (e.g. deserts).
This paper is structured as follows. In section 2, a description of the numerical model and simulations
conducted is given. The observational and reanalysis datasets considered and verification diagnostics used
to assess the WRF performance are also summarized. The meteorological conditions on 14 August 2013,
the event targeted in this work, are analysed in section 3. In section 4, the results of the model simulations
are discussed, with the main findings of the study outlined in section 5.





## 2. Model, Datasets and Diagnostics

### 2.1 Numerical Model

The numerical model used in this study is the WRF model version 4.2.1 (Skamarock et al., 2019). WRF is a fully compressible, non-hydrostatic, community model, which makes use of the Arakawa-C grid staggering for horizontal discretization and employs the Lorenz grid for vertical discretization. In all simulations WRF is initialized on 13 August 2013 and run for 48 h, with the first 24 h discarded as model spin-up. As discussed in section 3, the 14 August 2013 convective event is selected as it features both deep convection and a dusty atmosphere over the UAE. The initial and boundary conditions are taken from ERA-5 data (Hersbach et al., 2020), the latest reanalysis dataset of the European Center for Medium Range Weather Forecasts, which provides meteorological fields on a 0.25º × 0.25º grid and on an hourly basis, from 1979 to present. WRF is run in a three-nest configuration, with the spatial extent of the model grids presented in Fig. 1a. The outermost grid is at a resolution of 22.5 km, and covers the vast majority of the Arabian Peninsula and surrounding region, while the innermost nest, at 2.5 km resolution, is centered over the UAE and extends into the adjacent Arabian Gulf and Sea of Oman (Fig. 1b). The boundary conditions from ERA-5 are relaxed on a five grid-point buffer zone (not displayed in Figs. 1a-b).

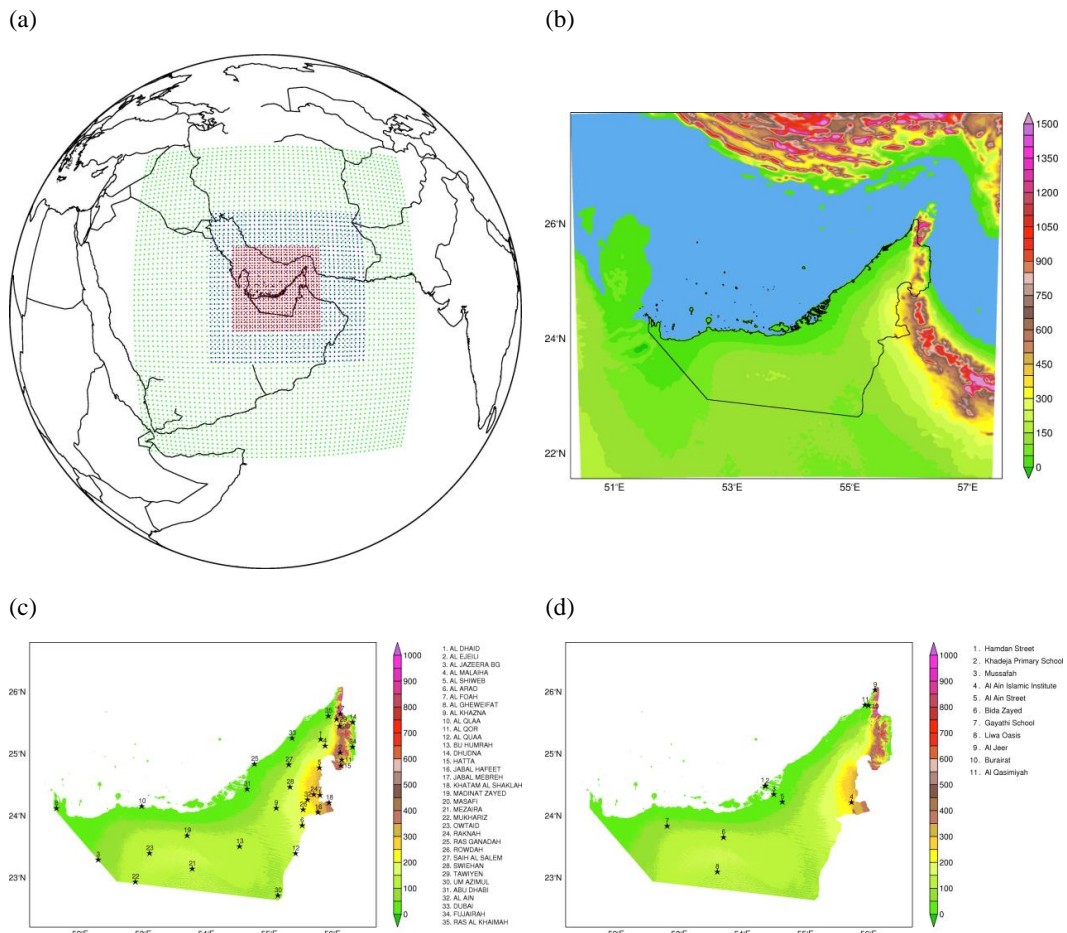

**Figure 1:** (a) Spatial extent of the WRF's 22.5 km (green), 7.5 km (blue) and 2.5 km (red) grids, used in the experiments. (b) Zoom-in view of the 2.5 km grid, with the shading giving the orography (m). (c) location and name of the NCM 30 automatic weather stations (#1-30) and 5 airport stations (#31-35) for which weather measurements are available on 14 August 2013. The orography is taken from a 30 m digital elevation model (Hulley et al., 2015). (d) is as (c) but for 11 NCM-EAD monitoring stations (stations #1-8 are maintained by EAD and #9-11 by NCM).


The physics schemes employed in the WRF simulations are summarized in Table 1. The model set up
reflects the findings of Schwitalla et al. (2020), who tested different WRF configurations for 14 July 2015
convective event in the UAE. Schwitalla et al. (2020) noted that a 0.025° grid (~2.7 km) may still be too
coarse to represent shallow clouds, and hence they employed a shallow cumulus scheme in their runs. The
same applies to the 2.5 km grid considered here, and for that purpose the mass-flux scheme embedded in



the MYNN PBL scheme, which parametrizes the non-convective component of the subgrid clouds (Olson
et al., 2019), was activated. As a result, no shallow cumulus scheme has to be employed in the 2.5 km grid.
The Noah-MP is configured following Weston et al. (2018) and Francis et al. (2021), while the sea surface
skin temperature scheme of Zeng and Beljaars (2005), which allows for the simulation of its diurnal cycle
and feedback on the atmosphere, is switched on. In the vertical, 45 levels are considered, more closely
spaced in the PBL, with the first level at about 27 m above the ground, and with the model top at 50 hPa.
Rayleigh damping is applied in the top 5 km to the wind components and potential temperature and on a
time-scale of 5 s to damp vertically-propagating waves (Skamarock et al., 2019). In all simulations, the
more realistic representation of the soil texture and land use land cover over the UAE described in Temimi
et al. (2020) is employed.

| Parameterization Scheme | Option |
|---|---|
| **Cloud Microphysics** | Thompson-Eidhammer scheme (Thompson and Eidhammer, 2014) *[In the default version, only ACI effects are activated; ARI effects are switched on through an option in the model's namelist]* |
| **Planetary Boundary Layer (PBL)** | Mellor-Yamada Nakanishi Niino (MYNN) level 2.5 (Nakanishi and Niino, 2006, 2009), with mass-flux scheme (Olson et al., 2019) activated |
| **Radiation** | Rapid Radiative Transfer Model for Global Circulation Models (Iacono et al., 2008) |
| **Cumulus** | *22.5 km and 7.5 km grids*: Kain-Fritsch (Kain, 2004), with subgrid-scale cloud feedbacks to radiation (Alapaty et al., 2012) *2.5 km grid*: no cumulus scheme |
| **Land Surface Model (LSM)** | Noah LSM with MultiParameterization options (Niu et al., 2011; Yang et al., 2011) |
| **Sea Surface Temperature (SST)** | 6-hourly ERA-5 SSTs + simple skin temperature scheme (Zeng and Beljaars, 2005) |

**Table 1:** Physics schemes employed in the WRF simulations







## 2.2. WRF Experiments

A total of ten WRF simulations were performed, listed in Table 2. The main difference between them is in the set-up of the Thompson-Eidhammer cloud microphysics scheme. This scheme, also known as Thompson aerosol-aware, is a modified version of the original Thompson scheme (Thompson et al., 2004, 2008), incorporating the activation of aerosols as cloud condensation nuclei and ice nuclei in a simplified manner (Thompson and Eidhammer, 2014). Two new variables, representing the concentration of hygroscopic or "water friendly" aerosols ($N_{wfa}$; designed to account for a combination of sulfates, sea salts, and organic matter) and non-hygroscopic or "ice friendly" aerosols ($N_{ifa}$; mineral dust), are added to the model. Aerosol direct and semi-direct effects (scattering and absorption of radiation; e.g. Spyrou et al., 2018) as well as indirect effects (aerosol-cloud interactions; e.g. Takenamura et al., 2005) can be accounted for in a relatively computationally cheap way, when compared e.g. to the simplest set up of the WRF-Chem (Grell et al., 2005) as noted e.g. by Saide et al. (2016). It is important to note that in the default version of the scheme only ACI effects are activated, the ARI effects are switched on through an option in the model's namelist.

| Numerical Experiment | Model Set up |
|---|---|
| WRF-1 | Idealised Aerosol Profiles (IDEAL) |
| WRF-2 | IDEAL + Aerosol Radiation Interactions with Rural Model (ARI_R) |
| WRF-3 | Climatological Aerosol Profiles (CLIM) |
| WRF-4 | CLIM + ARI_R |
| WRF-5 | CLIM + ARI_R + Grid Nuding in Outermost Nest (OUTNUD) |
| WRF-6 | CLIM + ARI_R + Grid Nuding in Two Outermost Nests (TWONUD) |
| WRF-7 | CLIM scaled by a factor of 5 (5×CLIM) + ARI_R + TWONUD |
| WRF-8 | 5×CLIM + ARI with Urban Aerosol Model (ARI_U) + TWONUD |
| WRF-9 | 5×CLIM + ARI with Maritime Aerosol Model (ARI_M) + TWONUD |



| WRF-10 | CLIM scaled by a factor of 10 (10×CLIM) + ARI_R + TWONUD |
|--------|----------------------------------------------------------|

**Table 2:** List of the WRF simulations discussed in this study.


There are two ways to initialize the aerosol concentration arrays: (i) employ an idealised profile based
on prescribed concentrations and the terrain height (hereafter IDEAL); (ii) extract the aerosol profiles from
a 7-year (2001-2007) simulation with the Goddard Chemistry Aerosol Radiation and Transport (GOCART;
Ginoux et al., 2001) model, described in Colarco et al. (2010) (hereafter CLIM).

In (1), the aerosol concentration is defined a
$$N(z) = N_1 + N_0\, EXP\left[-\left(\frac{h(z) - h(1)}{1000}\right)N_3\right], \qquad (1)$$

with
$$N_3 = -\frac{1}{0.8}LOG\left(\frac{N_1}{N_0}\right) \; if \; h(1) \le 1000\,m$$

$$N_3 = -\frac{1}{0.01}LOG\left(\frac{N_1}{N_0}\right) \; if \; h(1) \ge 2500\,m$$

$$N_3 = -\frac{1}{0.8\,COS\,[h(1)\times 0.001 - 1]}LOG\left(\frac{N_1}{N_0}\right) \; if \; 1000\,m < h(1) < 2500\,m$$


In the equations above, $h(z)$ is the height of the model level $z$ in meters, with $h(1)$ being the height of the
first model level. The constants $N_1$ and $N_0$ are set to $50\times10^6\,m^3$ and $300\times10^6\,m^3$ for water-friendly aerosols,
and $0.5\times10^6$ m$^3$ and $1.5\times10^6$ m$^3$ for ice-friendly aerosols, respectively. This definition is based on the
premise that aerosols are mostly concentrated in the lowest part of the atmosphere, with a faster decrease
with height over the higher-terrain, and a profile tailored for the continental United States. Spatially
$N_{wfa}$ and $N_{ifa}$ are uniform at the start of the run, but evolve during the course of the model integration. In
(2), and as described in Thompson and Eidhammer (2014), a 0.5º × 1.25º dataset on a monthly time-scale
and on 30 vertical levels is downloaded from the model's website, comprising both water-friendly (sulfates,





sea salts and organic carbon) and ice-friendly (dust, with particle sizes larger than 0.5 μm) aerosols. As in
(1), $N_{wfa}$ and $N_{ifa}$ are advected and diffused as any other scalars over time. In experiments #7-9 the aerosol
loading was multiplied by a factor of 5 (5×CLIM) and in experiment #10 by a factor of 10 (10×CLIM), at
all levels and model grid-points, to further investigate the sensitivity of the model's response to the aerosol
loading.

In its default configuration, the water- and ice-friendly aerosols activate water droplets and ice crystals,
respectively, but their interactions with the radiation (i.e. scattering and absorption) are not accounted for.
In order to switch it on, assumptions have to be made regarding the aerosol properties, in particular the
single-scattering albedo, asymmetry factor and Angstrom exponent, which are also a function of the relative
humidity (RH) that determines the aerosol hygroscopicity. Three aerosol models are available in WRF:
rural, urban and maritime (Shettle and Fenn, 1979; Ruiz-Arias et al., 2014). The rural aerosol model
(ARI_R) is designed for cases where the contribution from urban and industrial sources is small. It assumes
a mixture of 70% water soluble (ammonium, calcium sulfate, organic compounds) and 30% dust-like
aerosols. The urban model (ARI_U) is a mixture of 80% rural aerosols and 20% carbonaceous (soot-like)
aerosols, which are assumed to have the same size distribution as both components of the rural model. As
a result of the soot-like particles, the aerosols will be more absorbing. In particular, the single-scattering
albedo, the ratio of the scattering to the extinction efficiency (a value of 1 indicates that all particle
extinction is due to scattering and a value of 0 that it is due to absorption), at the 440-625 nm band is in the
range 0.95 to 0.99 for RH values of 0% to 90% for the rural type, and in the range 0.64 to 0.94 for the urban
type (Hodzic and Duvel, 2018). The maritime aerosol model (ARI_M) also consists of two components:
sea-salt, and a continental component, assumed to be identical to the rural aerosol but with the very large
particles removed, as they will eventually fall out as the air mass moves across water. As a result, the
maritime aerosol model will be less absorbing than the default (rural) model. The sensitivity to the aerosol
model is explored in experiments #7-9.






277  In addition to differences in the initialization of the aerosol concentration and the choice of the aerosol

278 model to parameterize the aerosol effects, in some of the runs (or analysis) grid nudging (Staufer and

279 Seeman, 1990; Staufer et al., 1991) towards ERA-5 data is employed in the outermost (OUTNUD) or in

280 the two outermost (TWONUD) nests in an attempt to correct for the model's large-scale biases in the

281 innermost nest. In these runs, the horizontal wind components, water vapour mixing ratio and potential

282 temperature perturbation are nudged on a time-scale of 1 h above roughly 800 hPa excluding the PBL. This

283 nudging configuration is preferred so as to allow the model to develop its own structures while at the same

284 time constraining the atmospheric circulation in the free atmosphere (e.g. Wootten et al., 2016). The role

285 of nudging in the outer nests to the predictions of the innermost grid is explored in simulations #4-6.

286

### 287  2.3. Observational and Reanalysis Datasets

288  In order to evaluate the model performance, three in-situ and two satellite-derived datasets are used.

289 Station data collected by the National Center of Meteorology (NCM) is available at 30 automatic weather

290 stations (AWS) and 5 airport stations given in Fig. 1c. Air temperature, RH, sea-level pressure, and

291 horizontal wind direction and speed are available every 15 min at the former and 1 h at the latter on 14

292 August 2013, with the downward shortwave radiation flux at the surface also measured at the location of

293 the AWS. Daily accumulated precipitation is available for all 35 stations. At 11 sites in the UAE, given in

294 Fig. 1d, hourly air quality measurements of particulate matter with a diameter not exceeding 10 μm ($PM_{10}$),

295 collected by the Environmental Agency - Abu Dhabi (EAD; https://www.adairquality.ae/; Teixido et al.,

296 2020), stations #1-8, and the NCM, stations #9-11, are available for model evaluation. In addition to the

297 surface/near-surface measurements, the 00 and 12 UTC radiosonde profiles at Abu Dhabi's International

298 Airport (24.4331°N, 54.6511°E) from the National Oceanic and Atmospheric Administration Integrated

299 Radiosonde Archive (IGRA; Durre et al., 2016; Durre and Xungang, 2008) are considered.





The satellite-derived datasets comprise (i) Red-Green-Blue (RGB) satellite images obtained from the
Spinning Enhanced Visible and Infrared Imager (SEVIRI) instrument onboard the Meteosat Second
Generation spacecraft (Banks et al., 2019), and (ii) Infrared Brightness Temperature (IRBT) maps from a
combination European, Japanese and United States geostationary satellites provided by the National Center
for Environmental Prediction / Climate Prediction Center (Janowiak et al., 2017). RGB images are available
every 15 min on a 0.05º (~5.6 km) grid for the domain 60ºS-60ºN and 60ºW-60ºE on the European
Organisation for the Exploitation of Meteorological Satellites (https://eoportal.eumetsat.int/) website. The
IRBT maps are at 4 km spatial resolution and 30 min temporal resolution, available from 60ºS-60ºN at all
longitudes,  on  the  National  Aeronautic  and  Space  Administration's  EarthData  website
(https://disc.gsfc.nasa.gov/datasets/GPM_MERGIR_1/summary).
Besides the listed observational datasets, the Modern-Era Retrospective analysis for Research and
Applications version 2 (MERRA-2; Gelato et al., 2017) data is also considered in this work. MERRA-2
explicitly accounts for aerosols and their interactions with the climate system, and is used to assess the
spatial distribution of aerosols over the UAE on 14 August 2013. MERRA-2 provides aerosol-related
variables such as the AOD on a 0.625º × 0.5º global grid and on an hourly basis.
**2.4. Verification Diagnostics**
The performance of the WRF model is evaluated with the verification diagnostics proposed by Koh et
al. (2012). In particular, the model bias, normalised bias (μ), correlation (ρ), variance similarity (η), and
normalised error variance (α), defined in equations (3) to (7) below, are employed.

$$D = F - O, \quad (2)$$

$$BIAS \ = <D> = <F> - <O>, \quad (3)$$

$$\mu = \frac{<D>}{\sigma_D}, \quad (4)$$





$$\rho = \frac{1}{\sigma_O \sigma_F} < (\boldsymbol{F} - < \boldsymbol{F} >) \cdot (\boldsymbol{O} - < \boldsymbol{O} >) >, -1 \leq \rho \leq 1, \quad (5)$$

$$\eta = \frac{\sigma_O \sigma_F}{\frac{1}{2}\left(\sigma_O^2 + \sigma_F^2\right)}, 0 \leq \eta \leq 1, \quad (6)$$

$$\alpha = \frac{\sigma_D^2}{\sigma_O^2 + \sigma_F^2} \equiv 1 - \rho\eta, 0 \leq \alpha \leq 2, \quad (7)$$

In the equations above, $\boldsymbol{D}$ is the discrepancy between the model forecast $F$ and the observations $O$, while
$< X >$ and $\sigma_X$ are the mean and standard deviation of $X$, respectively.
The bias is defined as the mean discrepancy between the WRF and the observations, $< \boldsymbol{D} >$, while the
normalized bias is the ratio of the bias to the standard deviation of the discrepancy, $\sigma_D$. The latter is used
to assess whether the model biases can be regarded as significant: as explained in Koh et al. (2012), if $|\mu| <$
$0.5$, the contribution of the bias to the Root-Mean-Square-Error is less than roughly 10%, and hence the
biases can be deemed as not significant. The correlation ($\rho$) and the normalised error variance ($\eta$) are a
measure of the phase and amplitude agreement between the observed and modelled signals, respectively,
with the two sources of error accounted for in the $\alpha$ diagnostic. For a random forecast based on the
climatological mean, $\rho = 0$ and hence $\alpha = 1$. Hence, a model prediction is considered as practically useful
if $\alpha < 1$. The $\rho$, $\eta$ and $\alpha$ diagnostics are non-dimensional quantities, symmetric with respect to the
observations and forecasts, and applicable to both scalar and vector variables, making them suitable to be
used in this work. Further details regarding the listed diagnostics can be found in Koh et al. (2012).





## 3. Description of the Event (14 August 2013)

On 14 August 2013, deep convection and a dusty environment were ubiquitous in the UAE, as seen in Fig. 2. The RGB and IRTB maps in the afternoon and evening hours, given in the first two rows, show a rapid flare-up of convection in the local early afternoon hours, which affected mostly western and central parts of the country. The IBRT values dropped to around 190 K indicating rather cold cloud tops, a sign of very deep convection (Reddy and Rao, 2018), with the thick high-level clouds shaded in brown in the RGB images. Such low values of IRBT are more typical of tropical convective activity, such as that seen in tropical disturbances (e.g. Evan et al., 2020), than the average summertime convection in the UAE (e.g. Branch et al., 2020). A second (but less intense) round of convection took place in the evening to nighttime hours, with isolated convective cells developing over eastern UAE and western Oman in early to mid-afternoon hours, when convection typically flares up here (Branch et al., 2020).

Besides the unstable environment, on this day the atmosphere was also rather dusty. The third row of Fig. 2 gives the AOD from MERRA-2 reanalysis data. Values in excess of two were seen over the western half of the UAE at 11 UTC (15 LT), decreasing during the afternoon and early evening hours. While these are not unusually high values for this region (e.g. Nelli et al., 2021), AODs higher than two are commonly seen during dust storms (e.g. Beegum et al., 2018). Some of the reduction in the AOD may be attributed to transport by the low-level circulation, but the fact that the dusty region overlaps at least partially with the convection region suggests that convection-aerosol interactions have likely taken place.

The 14 August 2013 event was chosen by manually inspecting hourly IRBT and MERRA-2 AOD images for the summertime (June to September) periods for which NCM and EAD data are available, and selecting the one where the deepest convection, as given by the lowest IRBT, and the dustier environment, as given the highest AOD, co-occurred in the UAE.



**Figure 2:** RGB satellite images derived from the measurements taken by the SEVIRI instrument over the southern Arabian Peninsula on 14 August 2013 at around (a) 11, (b) 14 and (c) 17 UTC. In the figures the magenta to pink shading denotes dust, while white regions are sandy areas. Thick high-level clouds are shaded in orange or brown, while thin high-level clouds are given in dark brown to black. Dry land is shaded in pale blue during daytime and pale green at night. (d)-(f) and (g)-(i) are as (a)-(c) but for the satellite-derived IRBT (K) and the AOD (non-dimensional) from MERRA-2 reanalysis, respectively.



Figs. 3 and 4 show the sea-level pressure, 2-meter water vapour mixing ratio, and low-level winds on

14 August 2013 from ERA-5 every 2 h from 08 UTC (12 LT) to 18 UTC (22 LT). The AHL is initially over

the UAE and surrounding region, but at 12 UTC it shifts westward, lying over western parts of the country

and extending into Saudi Arabia and Qatar, where the minimum sea-level pressure lies. The clockwise

circulation around the AHL converges with the daytime sea-breeze from the Arabian Gulf. This

convergence is more evident around 12-14 UTC (16-18 LT), Figs. 3c-d, over central and western parts of

the country, around the time when the convection flared up rapidly (Figs. 2a-b, d-e), and weakened after 16

UTC (20 LT), Fig. 3e, when both the AHL and the sea-breeze faded away. The convective clouds that

developed over eastern UAE were likely triggered by the convergence of the AHL circulation with the sea-

breeze from the Sea of Oman and topographically-driven flows (cf. Fig. 3c-d with Fig. 2d), in line with the

findings of Schwitalla et al. (2020) and Francis et al. (2021). Fig. 4 shows that the near-surface air was

rather moist over the country on this day, with water vapour mixing ratios typically in the range 15-20 g kg$^{-}$

$^{1}$. Together with the low-level wind convergence, the large-scale environment was suitable for the

occurrence of deep convection in the region. A comparison of the satellite images, Figs. 3a-f, with the ITD

drawn as a solid white line in the panels of Fig. 4, reveals that, at least on this day, the clouds tended to

develop around this convergence line. It is interesting to note that the ITD on this day reached southern

parts of Iran to the north of the UAE, a behaviour that is expected in the warmer months: as explained in

Fonseca et al. (2021), the inland moistening by the sea-breezes from the Arabian Gulf, Sea of Oman and

Arabian Sea allows the 15°C isoline of dewpoint temperature, the metric used to diagnose the position of

the ITD, to propagate northwards into the Arabian Gulf, as seen in Fig. 4.

A comparison of the AOD plots given in Figs. 2g-i with the 10-meter horizontal wind vectors plotted in

Fig. 3, indicates that the accumulation of aerosols over western UAE is related to the presence of a closed

atmospheric circulation associated with the AHL in the region. The decreasing values of AOD in the

evening to nighttime hours, are likely due to the advection of cleaner air from the south (cf. Figs. 3e-f), as

well as due to the washout and clearing of the air after the occurrence of precipitation in the region. As far



as the mechanism responsible for the dust emission is concerned, two factors are at play: (i) dust lifted by
strong near-surface winds triggered by cold pools and downbursts in association with the deep convection
that developed on this day, a well-known mechanism for dust lifting in arid regions (e.g. Cuesta et al., 2009;
Bou Karam et al., 2014; Francis et al., 2019a); (ii) strong southerly winds in the early morning, from the
combined effect of the AHL and sea-land breeze circulations, with the low-level wind convergence along
the ITD (Figs. 4a-b) aiding in the dust lifting activities by high turbulent winds at the leading edge of the
ITD (Bou Karam et al., 2008; 2009).

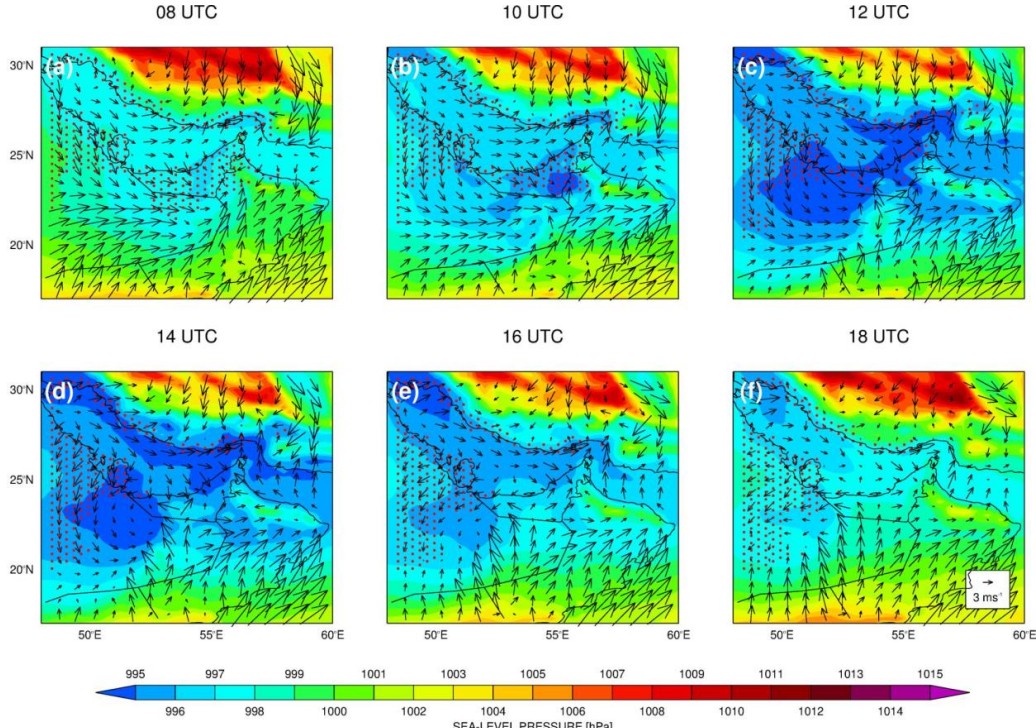

**Figure 3:** Sea-level pressure (shading; hPa) and 10-m horizontal wind vectors (arrows; m s$^{-1}$) at (a) 08 UTC, (b) 10 UTC, (c) 12 UTC, (d) 14 UTC, (e) 16 UTC and (f) 18 UTC on 14 August 2013 from ERA-5 data. The dotted region gives the AHL, defined based on the low-level atmospheric thickness (700-925 hPa), following Fonseca et al. (2021).




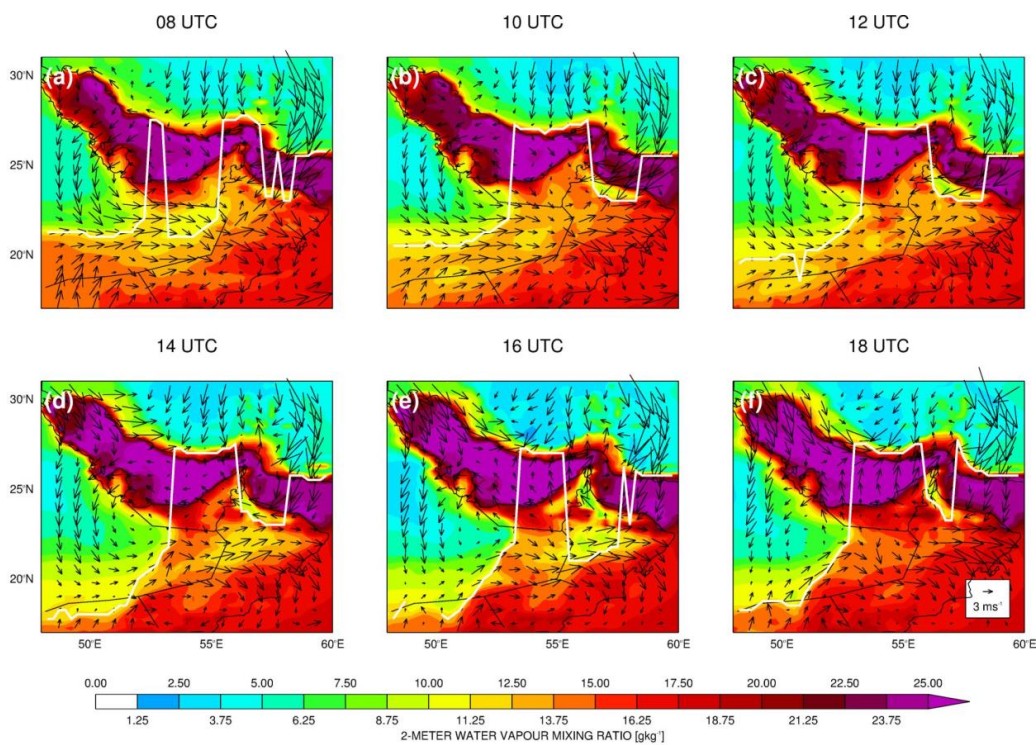

**Figure 4**: As Fig. 3 but with the 2-meter water vapour mixing ratio (g kg$^{-1}$) in shading, and the 850 hPa horizontal wind vector (m s$^{-1}$) in arrows. The solid white line gives the ITD defined using the 15°C isoline of dewpoint temperature (isodrosotherm).






## 4. WRF Simulations

### 4.1 Aerosol Loading




Figs. 5a-b show the concentration of water- and ice-friendly aerosols in the lowest model layer for the
simulations with the idealised (WRF-1) and the climatological (WRF-3) aerosol distribution. The main
difference between the two is in the order of magnitude, with roughly 10 times more aerosols in WRF-3
compared to WRF-1. This is not surprising: as stated in section 2.2, the idealised distribution was designed
for continental United States, where the atmosphere is cleaner compared to that in the UAE and surrounding
region. In fact, over India, and during the summer monsoon, the observed aerosol loading within the
boundary layer, as measured at the surface and by aircrafts, was found to be roughly 10 times larger than
that employed in the idealised profiles in WRF (Sarangi et al., 2018). The spatially uniform aerosol loading
at the start of the run in Fig. 5a, in line with the way it is coded in the model, contrasts with a heterogeneous
pattern in the simulation forced with a 7-year climatological aerosol loading. The higher amount of water-
friendly aerosols (sulfates, sea salt, organic matter) over the Arabian Gulf and of ice-friendly aerosols
(mineral dust) over inland areas in Saudi Arabia and Oman is consistent with the fact that the former are
typically advected from industrial and urban sites as well as from water bodies by the background
northwesterly winds, while the latter has its source in the Rub' Al Khali desert (e.g. Nelli et al., 2021).
Despite differences in the initialization and order of magnitude, the spatial pattern of aerosol loading is
similar in the two configurations, with a marked northwest - southeast gradient over the UAE. This can be
explained by the near-surface circulation, given in Fig. 6a for WRF-3 (similar results are obtained for WRF-
1, not shown). A comparison with Fig. 6b, same fields but from ERA-5, reveals that the AHL in WRF, at
12 UTC and as given by the sea-level pressure, is broader and displaced to the southeast with respect to that
in ERA-5. The associated cyclonic circulation acts to slow down the progression of the sea-breeze over
central and eastern parts of the country, where the model is drier than the reanalysis dataset, and speed it
up over western UAE, where it is more moist as the daytime sea-breeze is reinforced by the AHL





circulation. This explains why, in Fig. 1, the higher aerosol concentrations over the Gulf extend well inland
in the western half of the country, but are confined to coastal areas elsewhere.

Figs. 5c-d give the vertically averaged profiles over the UAE at 00 and 12 UTC for both WRF-1 and WRF-
3 simulations. The decrease in aerosol concentration with height is more pronounced in the runs with the
climatological profile, and in particular for the ice-friendly aerosols. This is consistent with the fact that
dust is primarily present at low elevations as its source is surface emissions in semi-arid/arid regions (Nelli
et al., 2021), whereas other aerosol types have more varied sources and are more ubiquitous in the
troposphere. The diurnal variability is small except at low elevations, below 700 hPa, where the well-mixed
daytime boundary layer leads to approximately constant values whereas at night, the concentrations are
higher just above the surface, as the aerosols are trapped below the low-level nighttime surface-based
inversion, and in the residual mixed layer above it. This variability is in line with the findings of Filioglou
et al. (2020) and Nelli et al. (2021). The aerosol concentration profile shown in Figs. 5c-d resembles the
observed profiles measured during dedicated field campaigns (e.g. Varghese et al., 2021).













(a)  (b)

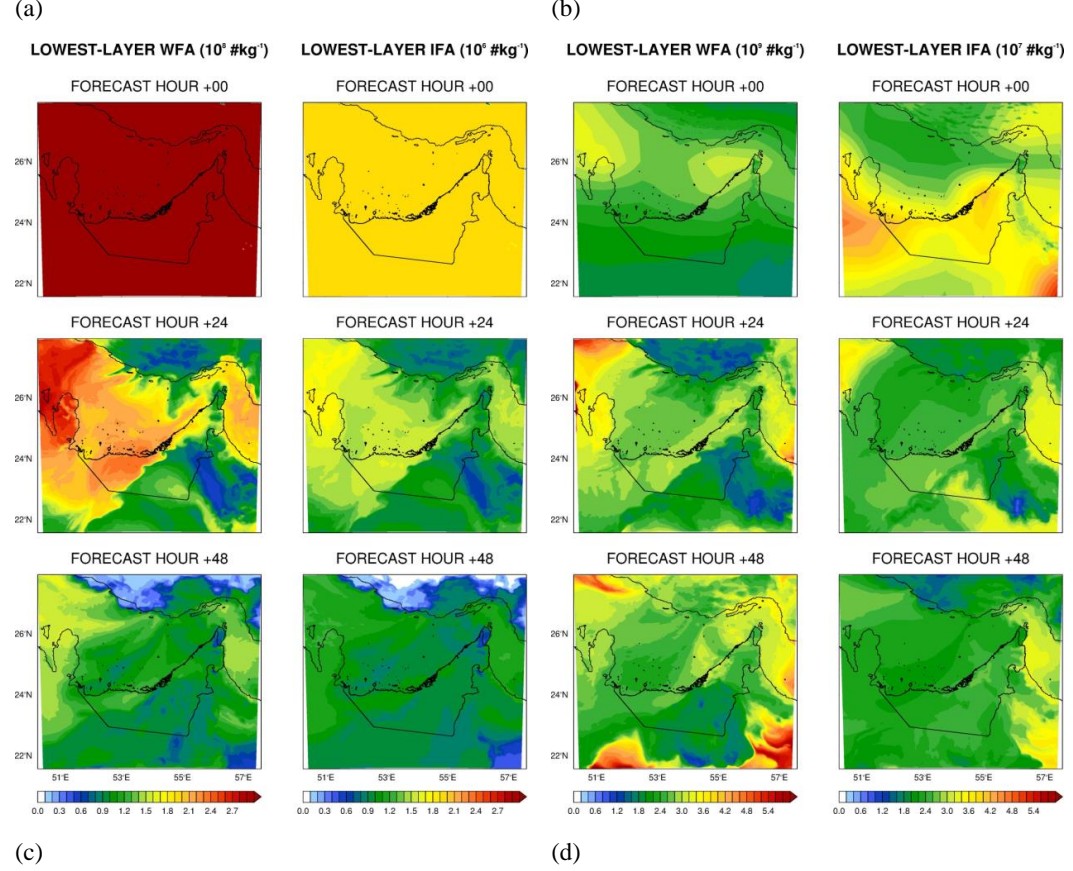

(c)  (d)

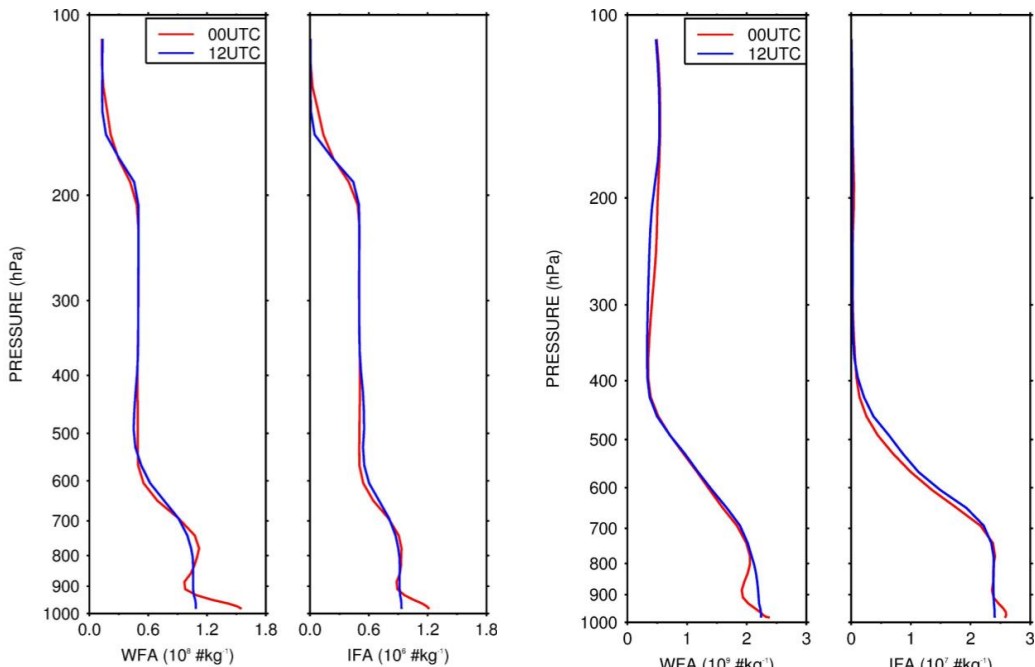

**Figure 5**: Concentration (#kg$^{-1}$) of water- and ice-friendly aerosols in lowest WRF layer at the start of the run (13 August at 00 UTC), and after 24 h (14 August at 00 UTC) and 48 h (15 August at 00 UTC) for the (a) WRF-1 and (b) WRF-3 simulations. The concentrations are divided by 10$^8$ and 10$^9$ for the water-friendly and by 10$^6$ and 10$^7$ for the ice-friendly aerosols for simulations WRF-1 and WRF-3, respectively. The fields are shown for the innermost (2.5 km) WRF grid. UAE-averaged vertical profiles of the water-friendly (left) and ice-friendly (right) aerosol concentration at 00 UTC (red) and 12 UTC (blue) on 14 August 2013 for the (c) WRF-1 and (d) WRF-3 simulations. The aerosol concentration in panels (c) and (d) is scaled as in panels (a) and (b), respectively.












(a)

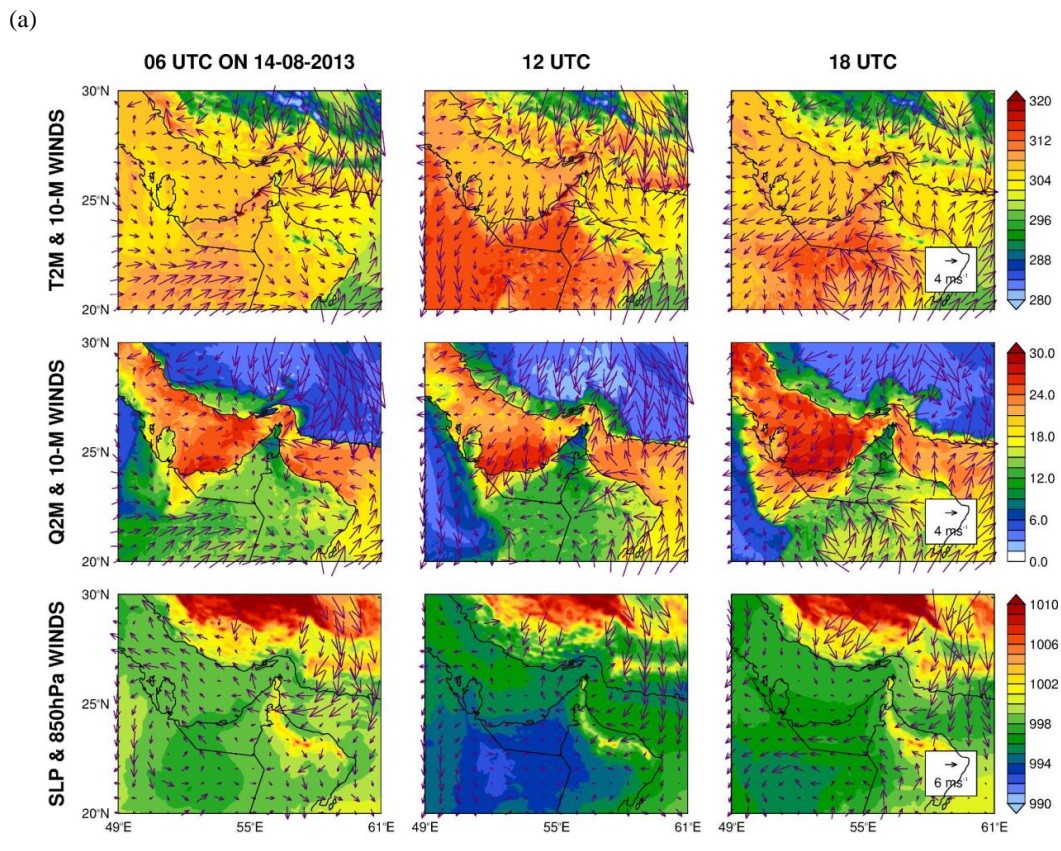

(b)



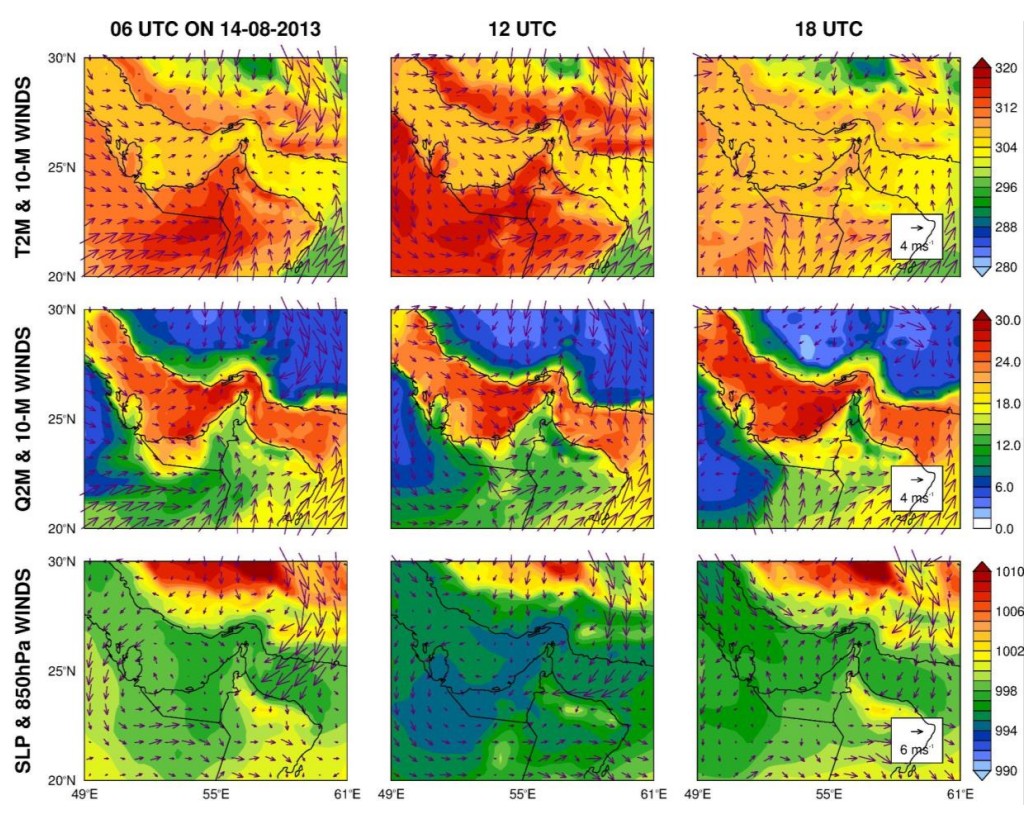

(c)


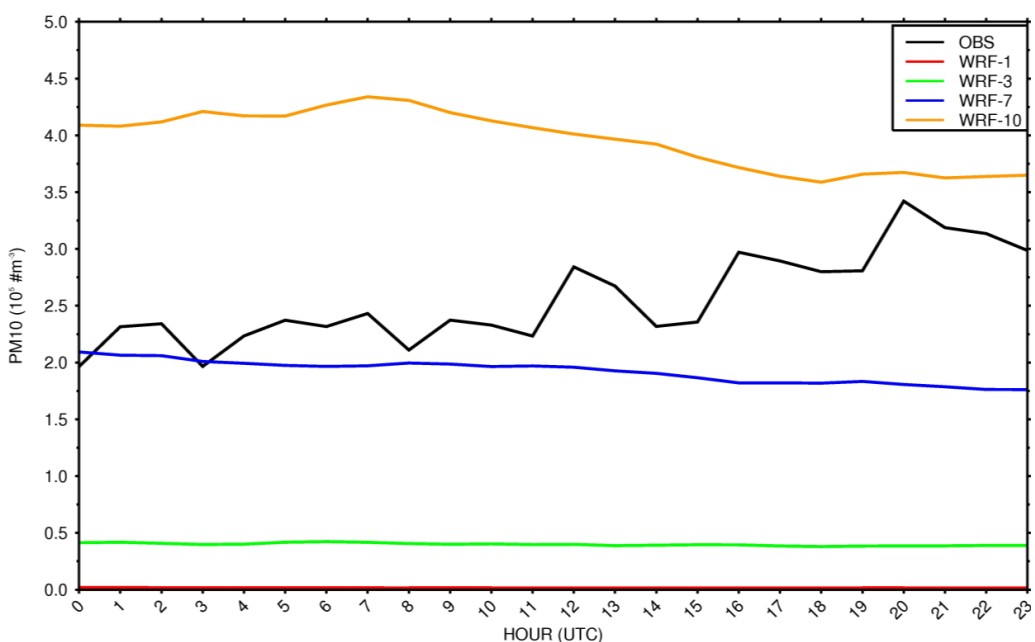

(d)

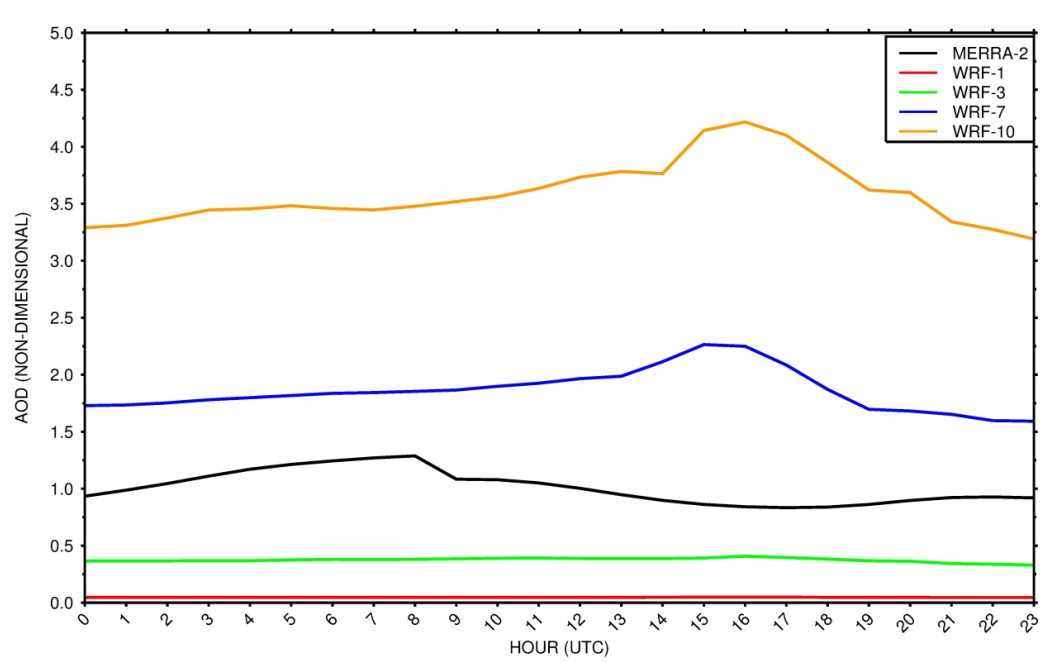


**Figure 6**: (a) 2-meter air temperature (K) and 10-meter wind vector ($m\,s^{-1}$) (top row), 2-meter water vapour mixing ratio ($g\,kg^{-1}$) and 10-meter wind vector ($m\,s^{-1}$) (middle row), and sea-level pressure (hPa) and 850 hPa wind vector ($m\,s^{-1}$) (bottom row) on 14 August 2013 at 06, 12 and 18 UTC for the 7.5 km grid of the WRF-3 simulation. (b) is as (a) but for ERA-5 reanalysis data. (c) Observed (black) and model-predicted for the 2.5 km grid of WRF-1 (red), WRF-3 (green), WRF-7 (blue) and WRF-10 (orange) simulations PM10 ($10^5\ \#\,m^{-3}$) averaged over all 12 stations in Fig. 1d. (d) UAE-averaged AOD from MERRA-2 (black) and the four WRF simulations shown in panel (c).



465  Given the marked difference in aerosol loading between the runs with the idealised and climatological

466  aerosol distributions, it is important to assess which is closer to the actual observed values. In order to do

467  this, the observed PM10 at the location of the 11 stations in Fig. 1d on 14 August 2013 was averaged and

468  compared with that given by WRF, Fig. 6c. Some approximations were made: e.g. the observed

469  measurements given in $\mu g\,m^{-3}$ and the WRF concentrations expressed in $\#\,kg^{-1}$ are converted to $\#\,m^{-3}$ using

470  a density of 1.65 $g\,cm^{-3}$; the WRF ice-friendly aerosol concentration in the lowest model layer at the closest

471  grid-point to the location of a station is directly compared with the observations at that station, assuming a

472  uniform loading in the model gridbox. As can be seen, even in the simulation with the climatological

473  distribution, there are fewer aerosols in the model with respect to those observed. A better agreement is

474  obtained when the latter is scaled by a factor of 5, which is done in runs WRF-7 to WRF-9, with an order

475  of magnitude increase, WRF-10, leading to a dustier environment in WRF. Besides the magnitude, the

476  downward trend in WRF, also seen in Fig. 5, contrasts with the increasing trend in the observations during

477  14 August 2013. An inspection of the trend for the individual stations revealed that the upward tendency is

478  mostly seen at stations #1-3 and #5 located around Abu Dhabi. It is then possible that the incorrect

479  representation of the observed low-level circulation by the model (cf. Fig. 6a with Fig. 6b), in particular

480  with respect to the position of the AHL / ITD earlier in the day, and the occurrence of precipitation and

481  associated cold pools later in the day (as noted in section 3, both factors played an important role in the dust

482  lifting activities on this day), may explain the opposite diurnal tendencies. Besides the aerosol concentration

483  just above the surface, it is also of interest to compare the model-predicted and observed AOD, which is a

484  column integral and gives information on the attenuation of the incoming solar radiation as it goes through

485  the atmosphere. Due to the extensive cloud cover on this day, Figs. 2a-c, the observed AOD from ground-



based and satellite assets exhibit gaps and missing data and therefore are not suitable to be used here (not
shown). Hence, the WRF-predicted AOD is compared with that of MERRA-2 reanalysis data in Fig. 6d.
The WRF-7 simulation, for which the climatological aerosol distribution is multiplied by a factor of 5, gives
the best agreement with the MERRA-2 AOD out of all model configurations considered. However, even in
this simulation the atmosphere in WRF it is slightly dustier, in particular in the afternoon hours, likely due
to a lack of precipitation that precludes a washout of the aerosols and a cleaning of the air, as discussed in
the next section. In any case, it is important to note that, despite the data assimilation, MERRA-2 still has
biases when compared to observed measurements, mostly due to missing emissions and/or deficiencies in
the parameterization schemes, as noted in Buchard et al. (2017). In addition, the upward trend in WRF's
AOD contrasts with the downward trend in the surface aerosol concentration on this day (cf. Figs. 6c and
6d). The AOD is estimated from the aerosol concentration, air density, thickness and RH of each model
layer (Thompson and Eidhammer, 2014). While the first two fields are clearly higher in the lower-
troposphere when compared to the upper-troposphere (e.g. see Figs. 5c-d for the aerosol concentration), the
opposite for the layer thickness that generally increases with height, the RH has considerable variability in
the column (e.g. Fig. A1b). As a result, the two aforementioned aerosol-related quantities do not have a
trend of the same sign.
**4.2 Aerosols Interaction with Convection**
**4.2.1 ARI on Idealised and Climatological Aerosol Distributions (WRF-1 to WRF-4)**
In order to investigate the impact of switching on the ARI on the simulations with the idealised and
climatological aerosol distributions, Fig. 7 shows the WRF bias, with respect to hourly station data, for air
temperature, water vapour mixing ratio, horizontal wind speed and surface downward shortwave radiation
flux, averaged over all 35 NCM stations on 14 August 2013. The scores averaged over all hours of the day
are given in Table 3.





As expected, when the ARI is switched on, there is a decrease in the shortwave radiation flux reaching the
surface, Fig. 7d, which is more pronounced for the run with the climatological distribution owing to the
higher aerosol loading. Compared to the simulations where the ARI is switched off, the maximum reduction
in the radiation flux is ~10 W m$^{-2}$ for the run with the idealised aerosol distribution and ~40 W m$^{-2}$ for the
run with the climatological aerosol distribution, with daily-averaged values of 3 W m$^{-2}$ and 20 W m$^{-2}$,
respectively. Despite the small decrease in the the downward shortwave radiation flux, however, WRF
continues to largely overestimate the observed values, which can be attributed to a lack of clouds in the
model, a bias that has been noted by several authors (e.g. Wehbe et al., 2019; Fonseca et al., 2020; Temimi
et al., 2020). Given the lack of clouds, the ARI effects will prevail over the ACI effects, and hence the
model predictions for simulations WRF-1 to WRF-4 will be comparable, as seen in Fig. 7, as the radiative
impacts of switching on the ARI are small. This can be seen in fields like the air and surface temperatures,
for which the decreases are within 0.5 K and 1 K, respectively, when the ARI effects are activated. These
changes are comparable to those reported by other authors for a similar variation of the surface radiation
fluxes (e.g. Sun et al., 2012; Menut et al., 2019).

In all simulations, WRF is much colder than observations, with biases of up to 7 K and on a daily-average
around 2.5 K. This has been reported in the literature (e.g. Weston et al., 2018; Temimi et al., 2020), with
the biases more pronounced in the warmer months and not being restricted to the Arabian Desert (e.g. Fekih
and Mohamed, 2019). They may arise from deficiencies in the physical parameterization schemes, in
particular in the LSM and radiation schemes, and/or an incorrect representation of the atmospheric
composition. Several attempts have been made to correct for this bias, such as employing different model
configurations (e.g. Chaouch et al., 2017; Schwitalla et al. 2020) and input data (e.g. Francis et al., 2021),
tuning hard-coded parameters (e.g. Weston et al., 2018; Nelli et al., 2020b), and using more realistic lower
boundary conditions (e.g. Temimi et al., 2020). The sensitivity experiments described in Fig. 7 suggest that
having a more realistic representation of the aerosol loading does not alleviate the cold bias either, with





differences within ±0.15 K for the daily-averaged air temperature (Table 3). It is then possible that the
referred cold bias could be down due to a non-linear interaction of different model errors.

Besides the cold temperatures, the near-surface wind speed is also too strong when compared to that
observed, Fig. 7c. The two biases can be related, as too strong turbulent mixing will lead to cooler and drier
near-surface conditions (Oke, 1988), the latter consistent with the negative mixing ratio biases of up to -4.5
$g\,kg^{-1}$ and on average around -2.2 $g\,kg^{-1}$ (Fig. 7b). The stronger near-surface winds in the model are likely
a result of an incorrect representation of its subgrid-scale fluctuations and deficiencies in the surface drag
parameterization, as optimizing relevant parameters such as the roughness length does not seem to alleviate
the problem (Nelli et al., 2020). Changing the aerosol loading by an order of magnitude only leads to
differences of up to ±0.2 $m\,s^{-1}$ in the daily-mean wind speed (Table 3), or less than 6% of the daily-averaged
values. In a nutshell, the major impact of switching on the ARI is a decrease in the downward shortwave
radiation flux, which reaches up to 40 $W\,m^{-2}$ when the more opaque climatological distribution is employed.
It is interesting to note that, for all fields given in Fig. 7, the magnitude of the WRF biases exceed that of
the response to the aerosol loading and to the activation of the ARI.



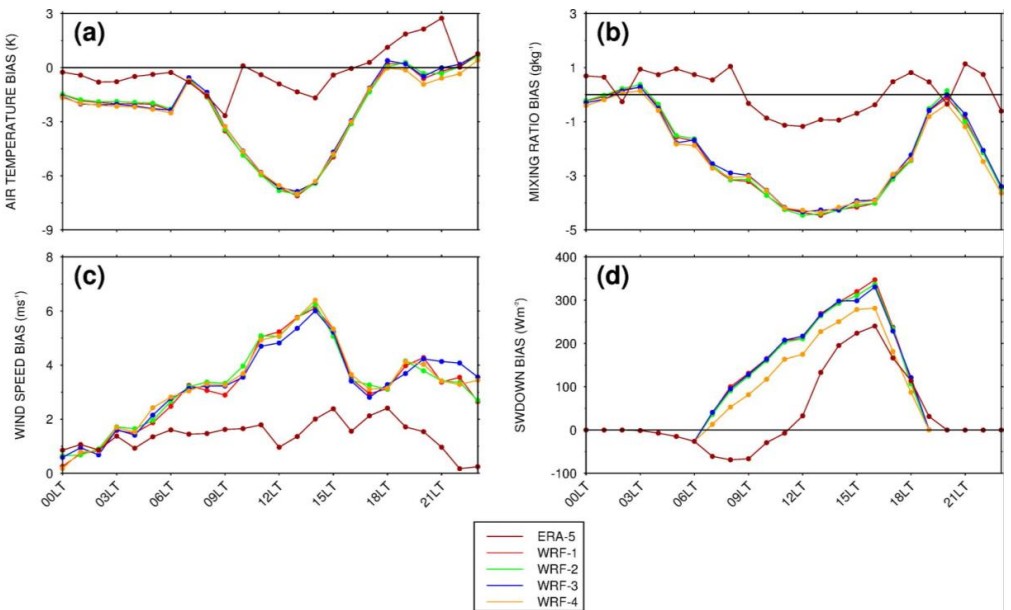

**Figure 7**: 2-meter (a) temperature (ºC) and (b) water vapour mixing ratio (g kg$^{-1}$), (c) 10-meter horizontal wind speed (m s$^{-1}$) and (d) surface downward shortwave radiation flux (W m$^{-2}$) bias with respect to in situ measurements averaged over the location of the 35 NCM stations given in Fig. 1c for simulations WRF-1 (red), WRF-2 (green), WRF-3 (blue) and WRF-4 (orange). The brown line gives the biases for ERA-5. The time in the horizontal axis is LT on 14 August 2013.



The verification diagnostics when all hours of the day and 35 weather stations are considered are given
in Table 3. In line with Fig. 7, the scores are roughly comparable for simulations WRF-1 to WRF-4. Except
for sea-level pressure, the α scores are always less than 1, indicating that the model predictions can be
regarded as skillful. For all variables shown, phase errors dominate over magnitude errors, as η is typically
larger than 0.95, while ρ is negative at times. A similar conclusion was reached by Fonseca et al. (2020),
in the analysis of a cold season and warm season convective events in the UAE. The lack of clouds and the
drier environment in the model will impact the diurnal cycle of variables such as air temperature and mixing
ratio, which exhibit higher α values when compared to the shortwave radiation flux, for which the diurnal
variability is rather well captured by WRF, with both ρ and η in excess of 0.9. The poorer scores for sea-
level pressure are consistent with the incorrect simulation of the AHL (cf. Figs. 6a and Fig. 3), both in terms





of its magnitude and temporal variability, while the lower values of ρ (and hence higher values of α) for
the wind vector are a reflection of its higher temporal and spatial variability, which are rather difficult to
model in the UAE, as noted by Fonseca et al. (2020) and Nelli et al. (2020b). Except for the water vapour
mixing ratio, the absolute value of the normalised bias is generally higher than 0.5 for WRF-1 to WRF-4,
meaning that the WRF tendency to underpredict the air temperature and overestimate the strength of the
near-surface wind, can be regarded as significant. Fig. A1 shows the bias in the temperature and RH profiles
at the location of Abu Dhabi's airport, and with respect to radiosonde data, at 00 and 12 UTC on this day.
In order to extract this quantity, first the observed and model-predicted data was interpolated in log-pressure
coordinates to a pre-defined set of pressure levels from 1000 to 100 hPa at which the observational data is
typically available, before the difference between each set of WRF and observed profiles was taken. The
WRF temperature biases are typically within ±2 K, having the largest amplitudes between 950 and 800 hPa
at 00 UTC. The magnitude of the biases decreases from a peak of about 3 K for WRF-2 to 1.5 K for WRF-
4, with the warming consistent with the increased dust loading (Figs. 5c-d). A smaller warming tendency
of up to 0.5 K is also seen when the ARI effects are switched on, in particular when the climatological
aerosol loading is used (WRF-3 vs. WRF-4). The temperature biases at 12 UTC have a lower magnitude
likely because of the well-mixed vertical profile in the lower layers, which leads to a roughly uniform
aerosol loading below roughly 700 hPa (Figs. 5c-d). The RH vertical profile in WRF is clearly drier than in
observations, in particular at 12 UTC, in line with the less moist near-surface environment. The tendency
of the model to generate drier conditions at the site in the summer season has been reported by Temimi et
al. (2020) over the UAE and Fountoukis et al. (2018) over Qatar. Besides deficiencies in the physics
schemes, the drier environment may be explained by a lack of clouds in WRF, which is consistent with the
reduced amounts of precipitation generated by the model (Table 3) and the cooler temperature profile (cf.
Figs. A1a-b), and has been found to be the case in summertime convective events in the region (e.g. Francis
et al., 2021).




| Field | Diagnostic | WRF-1 | WRF-2 | WRF-3 | WRF-4 | WRF-5 | WRF-6 | WRF-7 | WRF-8 | WRF-9 | WRF-10 |
|---|---|---|---|---|---|---|---|---|---|---|---|
| Temperature | BIAS (K) | -2.4720 | -2.4530 | -2.4050 | -2.5551 | -2.6212 | -2.5464 | -2.7312 | -3.4674 | -2.8168 | -3.0556 |
| | $\mu$ | -0.5263 | -0.5219 | -0.5166 | -0.5603 | -0.5790 | -0.5746 | -0.6649 | -1.0292 | -0.6428 | -0.7843 |
| | $\rho$ | 0.4113 | 0.4118 | 0.4255 | 0.4374 | 0.4400 | 0.4655 | 0.5213 | 0.6299 | 0.4815 | 0.5504 |
| | $\eta$ | 0.9979 | 0.9977 | 0.9975 | 0.9986 | 0.9990 | 0.9989 | 1.0000 | 0.9859 | 0.9997 | 0.9993 |
| | $\alpha$ | 0.5896 | 0.5892 | 0.5756 | 0.5632 | 0.5604 | 0.5350 | 0.4787 | 0.3790 | 0.5187 | 0.4499 |
| Mixing Ratio | BIAS (g kg$^{-1}$) | -2.2123 | -2.0726 | -2.4731 | -2.3181 | -2.4628 | -2.8098 | -2.6691 | -2.6477 | -2.8422 | -2.7686 |
| | $\mu$ | -0.3835 | -0.3605 | -0.4279 | -0.4004 | -0.4305 | -0.4731 | -0.4315 | -0.4205 | -0.4603 | -0.4170 |
| | $\rho$ | 0.3511 | 0.3563 | 0.3565 | 0.3399 | 0.3907 | 0.3942 | 0.3417 | 0.3341 | 0.3609 | 0.3041 |
| | $\eta$ | 0.9915 | 0.9916 | 0.9933 | 0.9898 | 0.9971 | 0.9999 | 1.0000 | 0.9995 | 0.9995 | 0.9962 |
| | $\alpha$ | 0.6519 | 0.6468 | 0.6459 | 0.6635 | 0.6105 | 0.6059 | 0.6584 | 0.6661 | 0.6393 | 0.6970 |
| SLP | BIAS (hPa) | 3.0872 | 3.0702 | 3.0680 | 3.0084 | 2.8557 | 2.7449 | 2.7320 | 2.6919 | 2.9786 | 2.8215 |
| | $\mu$ | 0.6995 | 0.6957 | 0.6940 | 0.6788 | 0.6500 | 0.6292 | 0.6231 | 0.6210 | 0.6823 | 0.6438 |
| | $\rho$ | -0.0456 | -0.0430 | -0.0442 | -0.0475 | -0.0603 | -0.0610 | -0.0734 | -0.0731 | -0.0809 | -0.0823 |
| | $\eta$ | 0.8324 | 0.8318 | 0.8310 | 0.8303 | 0.8387 | 0.8431 | 0.8431 | 0.8499 | 0.8474 | 0.8454 |
| | $\alpha$ | 1.0380 | 1.0358 | 1.0367 | 1.0394 | 1.0506 | 1.0515 | 1.0619 | 1.0621 | 1.0686 | 1.0696 |
| SWDOWN | BIAS (W m$^{-2}$) | 99.4563 | 96.7037 | 97.7780 | 77.5172 | 71.6176 | 73.7791 | 9.2294 | -112.3040 | 35.3777 | -45.8454 |
| | $\mu$ | 0.5863 | 0.5732 | 0.5717 | 0.4975 | 0.4678 | 0.4742 | 0.0747 | -0.5850 | 0.2613 | -0.3298 |
| | $\rho$ | 0.9082 | 0.9077 | 0.9059 | 0.9114 | 0.9126 | 0.9111 | 0.9182 | 0.8415 | 0.9118 | 0.9077 |
| | $\eta$ | 0.9736 | 0.9747 | 0.9738 | 0.9835 | 0.9850 | 0.9838 | 0.9995 | 0.8341 | 0.9982 | 0.9595 |
| | $\alpha$ | 0.1175 | 0.1152 | 0.1178 | 0.1036 | 0.1011 | 0.1036 | 0.0823 | 0.2981 | 0.0898 | 0.1291 |
| Horizontal Wind | BIAS (SPEED; m s$^{-1}$) | 3.0946 | 3.1309 | 3.1145 | 3.1785 | 3.2951 | 3.5708 | 4.1674 | 3.1660 | 4.0691 | 4.4585 |
| | $\mu$ (SPEED) | 0.7686 | 0.7572 | 0.7667 | 0.7572 | 0.7630 | 0.7817 | 0.8530 | 0.6813 | 0.8714 | 0.9156 |
| | $\rho$ | 0.1557 | 0.1571 | 0.1407 | 0.1226 | 0.1150 | 0.1293 | 0.0513 | 0.0785 | 0.0597 | 0.0182 |
| | $\eta$ | 0.9728 | 0.9679 | 0.9717 | 0.9715 | 0.9645 | 0.9568 | 0.9498 | 0.9252 | 0.9618 | 0.9545 |
| | $\alpha$ | 0.8485 | 0.8479 | 0.8633 | 0.8809 | 0.8891 | 0.8763 | 0.9513 | 0.9274 | 0.9425 | 0.9826 |
| PRECIPITATION BIAS (mm) | | -42.4447 | -40.5812 | -51.0678 | -50.4518 | -48.9050 | -38.0378 | -41.5867 | -35.7302 | -48.5239 | -45.6105 |

**Table 3:** Skill scores for air temperature, water vapour mixing ratio, sea-level pressure, downward shortwave radiation flux, horizontal wind vector and precipitation for all 35 NCM stations for the WRF simulations conducted in this study.


On this day, a total of 56.20 mm of rain was measured at all stations. However, the model biases for runs
WRF-1 to WRF-4 range between -42 and -51 mm, indicating that less than a quarter of the observed
precipitation is predicted by WRF. As seen in Figs. 8a-d and A2a-e, most of the rain and clouds develop to
the south of the UAE, due to a southward shift in the region of low-level wind convergence, as a result of
a broader and stronger AHL. This shift can be seen by comparing Figs. 6a and 6b: e.g. at 12 and 18 UTC,
in ERA-5, the low-level convergence is mostly over central UAE, while in WRF it is further south and
takes place later in the day, as the southerlies are weaker due to a more extensive thermal low. It is
interesting to note that using the climatological aerosol loading leads to slightly drier conditions at the
location of the NCM stations of 10-11 mm (Table 3), even though over the whole domain it rains more
(Figs. 8a-d) due to enhanced convection over northeastern Saudi Arabia (Figs. A2a-e). The reduction in
precipitation over the UAE in WRF-3 and WRF-4 compared to WRF-1 and WRF-2 may be attributed to
the drier conditions (Table 3), as well as to the stabilizing effect aerosols have on the environment, with a



heating of the aerosol layer and a cooling of the surface below (e.g. Guo and Yin, 2015), although aerosol-
precipitation effects are known to be highly sensitive to aerosol properties (Solmon et al., 2008). The drier
environment in WRF-3 and WRF-4 is mostly over western UAE, where there is additional precipitation in
WRF-1 and WRF-2, Figs. 8a-d, and is due to a late arrival of the sea-breeze that arises from a southeasterly
shift in the position of the AHL (not shown). The changes in the position and strength of the AHL with the
aerosol loading is discussed in more detail in subsection 4.2.3. Over the whole domain, however, WRF-4
is wetter than WRF-1 to WRF-3. In fact, while over the UAE the impact of switching on the ARI on the
model-predicted precipitation is rather small, generally less than 1 mm (Table 3), when the climatological
distribution is used it leads to a ~47% increase in the domain-wise rainfall (Figs. 8c-d). This arises from
deeper convection, as shown by the colder cloud tops in Figs. A2d-e as opposed to Figs. A2b-c, with the
stronger updrafts (Fig. A3) leading to a higher fraction of aerosols being activated (Thompson and
Eidhammer, 2014). Fig. A3 shows the maximum vertical velocity in the column, and the pressure level at
which it is predicted, in the model for runs WRF-3 and WRF-4. In the latter the vertical velocity has a larger
magnitude ($56\,\mathrm{m\,s^{-1}}$ vs. $31\,\mathrm{m\,s^{-1}}$), in both peaking at about $160\,\mathrm{hPa}$, a sign of overshooting convection (e.g.
Chaboureau et al., 2007). These findings are in line with the results of Huang et al. (2019), who found that
switching on the ARI effects delays the onset of convection due to the dust stabilizing effects, but leads to
more active cells later in the day, with an overall increase in rainfall.










(a) (b)

(c) (d)

(e) (f)



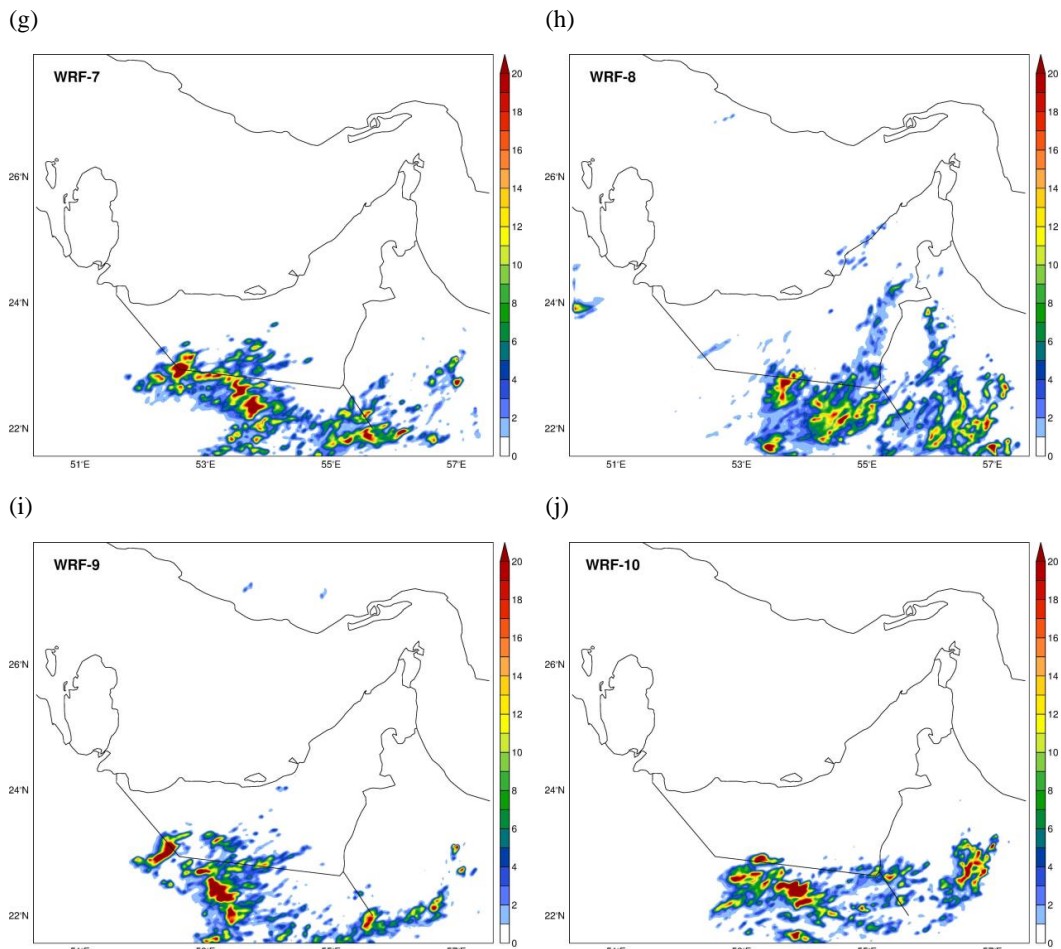

**Figure 8**: Total accumulated precipitation (mm) on 14 August 2013 for the 2.5 km WRF grid for simulations (a) WRF-1, (b) WRF-2, (c) WRF-3, (d) WRF-4, (e) WRF-5, (f) WRF-6, (g) WRF-7, (h) WRF-8, (i) WRF-9 and (j) WRF-10.



### 4.2.2 Impact of grid nudging on innermost nest predictions (WRF-4 to WRF-6)


As noted in the previous subsection, WRF has a considerable cold bias over this region, which is not
restricted to the UAE. However, when ERA-5 data, used to force the model, is compared with station data,
such cold bias is much reduced: it is mostly within 1 K and with a maximum value of 2.7 K, less than half





of the peak WRF bias (Fig. 7). As attempts to address this issue by modifying the WRF configuration have
not been successful (e.g. Chaouch et al., 2017; Nelli et al., 2020b; Temimi et al., 2020), interior nudging
towards ERA-5 was applied to the outermost and two outermost grids in an attempt to correct the
aforementioned model biases. As noted in section 2.2, the fields nudged include the water vapour mixing
ratio, temperature, and horizontal wind components above 800 hPa and on a time-scale of 1 h excluding the
PBL. Fig. 9 shows near-surface atmospheric fields for the run with the climatological aerosol loading and
without interior nudging, WRF-4, and their difference between the simulations with interior nudging (WRF-
5 and WRF-6) and this control run. The daily-averaged scores at the location of the 35 NCM stations are
given in Table 3.

When interior nudging is employed in the 22.5 km and 7.5 km grids, the model predictions in the 2.5 km
grid are generally more skillful when compared to the run where no interior nudging is applied or when it
is restricted to the 22.5 km grid (Table 3), as the output of the 7.5 km grid is directly used to generate
boundary conditions for the innermost nest. In particular, a comparison of Figs. 9a, 9c and 6b reveals that
the near-surface fields in the 2.5 km grid are corrected towards those in ERA-5, despite the fact that the
interior nudging is only applied above 800 hPa and in the outer grids. As an example, the air over central
and western UAE is more moist at 06 UTC and over the UAE it is generally warmer as well; the minimum
in sea-level pressure is shifted eastwards at this time, closer to that in ERA-5; at 12 and 18 UTC, the sea-
level pressures are higher in WRF-6 compared to WRF-4. These tendencies are also present in WRF-5 but
are of a smaller magnitude, as the ERA-5 signal is likely weakened by the lack of interior nudging in the
intermediate grid. These results are consistent with the findings of Wootten et al. (2016), who concluded
that employing analysis nudging in the interior of a 30 km and 10 km grids of a three-nest simulation, leads
to more accurate predictions in the 2 km innermost grid compared to when interior nudging is restricted to
the 30 km grid.



Table 3 shows that in WRF-6, the aforementioned cold bias is slightly reduced, albeit by only 0.01 K on a
daily-averaged scale. This is because WRF also generates more precipitation, Figs. 8d-f, which leads to
locally colder temperatures (cf. Fig. 9c). In both WRF-5 and WRF-6, the AHL is displaced to the east with
respect to WRF-4, in particular in the latter, with the low-level convergence of the associated cyclonic
circulation with the sea-breeze from the Arabian Gulf leading to increased rainfall over central and eastern
UAE, Figs. 8d-f. On the backside of the AHL, the enhanced moisture advection from the Arabian Gulf
augments the precipitation over southwestern UAE and adjacent Saudi Arabia, as evidenced by the deeper
convection in the region (Figs. A2e-g). Over northeastern UAE, on the other hand, the southeasterly winds
from the AHL bring in drier air and weaken the moistening effect of the sea breeze from the Sea of Oman
and Arabian Gulf, leading to a reduction of the 2-meter water vapour mixing ratio in excess of 10 $g\,kg^{-1}$ at
some sites in WRF-6. As a result, the averaged bias of this field at the location of the NCM stations increases
slightly from -2.32 $g\,kg^{-1}$ in simulation WRF-4, to -2.46 $g\,kg^{-1}$ in WRF-5 and -2.81 $g\,kg^{-1}$ in WRF-6. The air
temperature, sea-level pressure, downward shortwave radiation and precipitation scores, on the other hand,
are higher for WRF-6 compared to WRF-4 (Table 3). A marginal improvement is also seen in the vertical
profiles of temperature and RH with respect to the Abu Dhabi sounding data (Fig. A1). With respect to
WRF-4 (light blue curve), and in particular in WRF-6 (light green curve), there is a slight reduction of the
WRF biases: e.g. note the decrease in the air temperature biases around 500 hPa and 850-950 hPa at 00 UTC
and between 150 and 350 hPa at 12 UTC by up to 1 K, and in the RH biases between 550 and 700 hPa at12
UTC by up to 10 %.

In summary, while the application of interior nudging in the outermost or two outermost grids generally
improves the model performance, in line with the findings of other studies (e.g. Gomez-Navarro et al.,
2015; Wotten et al., 2016), in some regions (e.g. northeastern UAE) it may have detrimental effects, due to
its impact on the atmospheric circulation. Nevertheless, simulation WRF-6 is preferred to WRF-4 and
WRF-5, as per the scores given in Table 3, and will be selected as reference for the sensitivity study on the
aerosol loading and properties discussed in the next subsections.




(a)

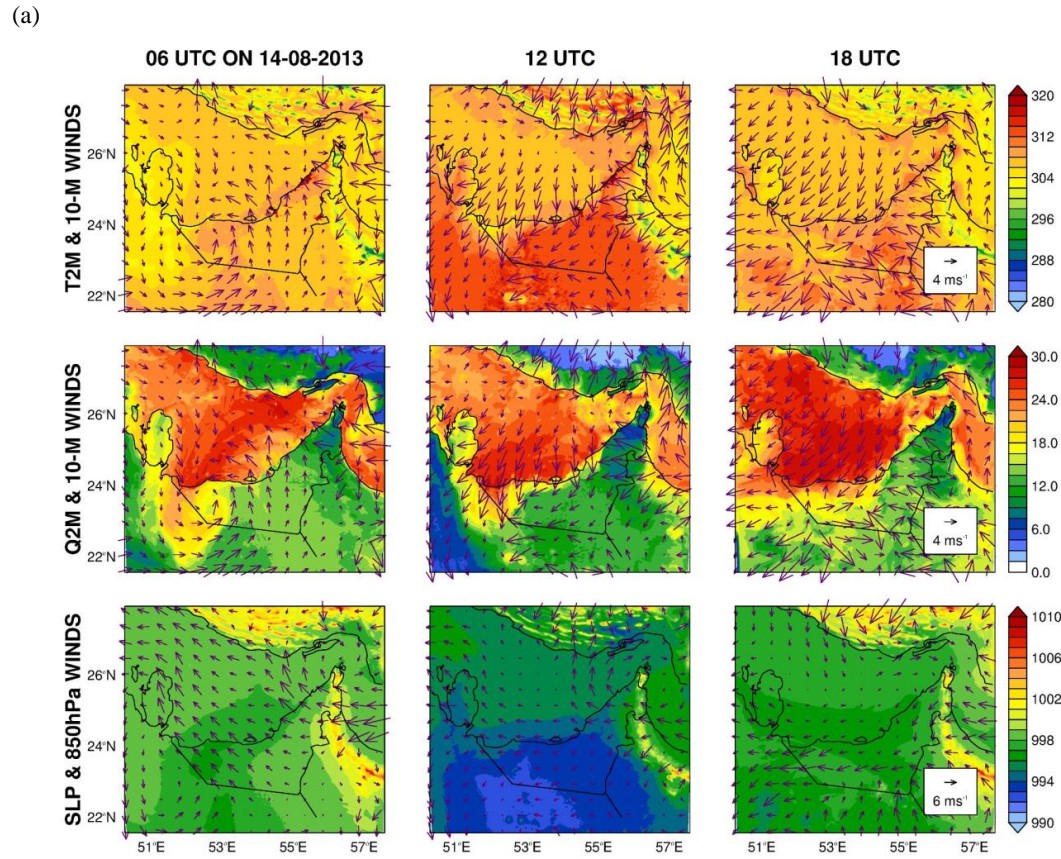

(b)



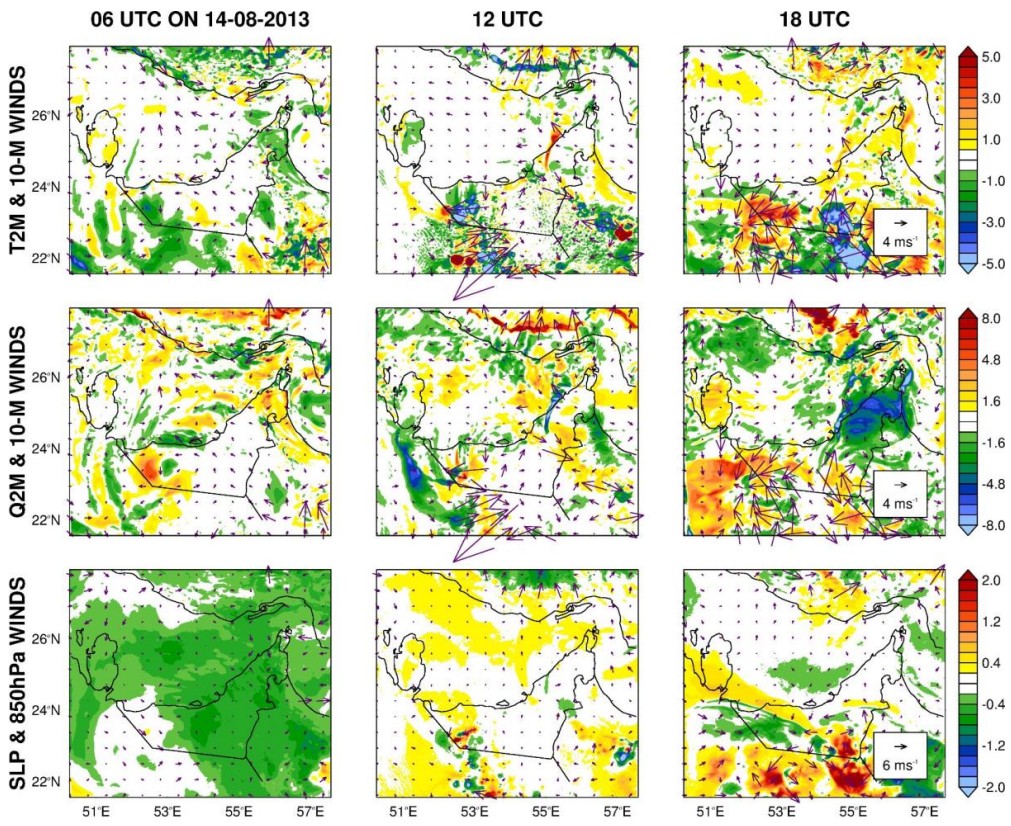

(c)

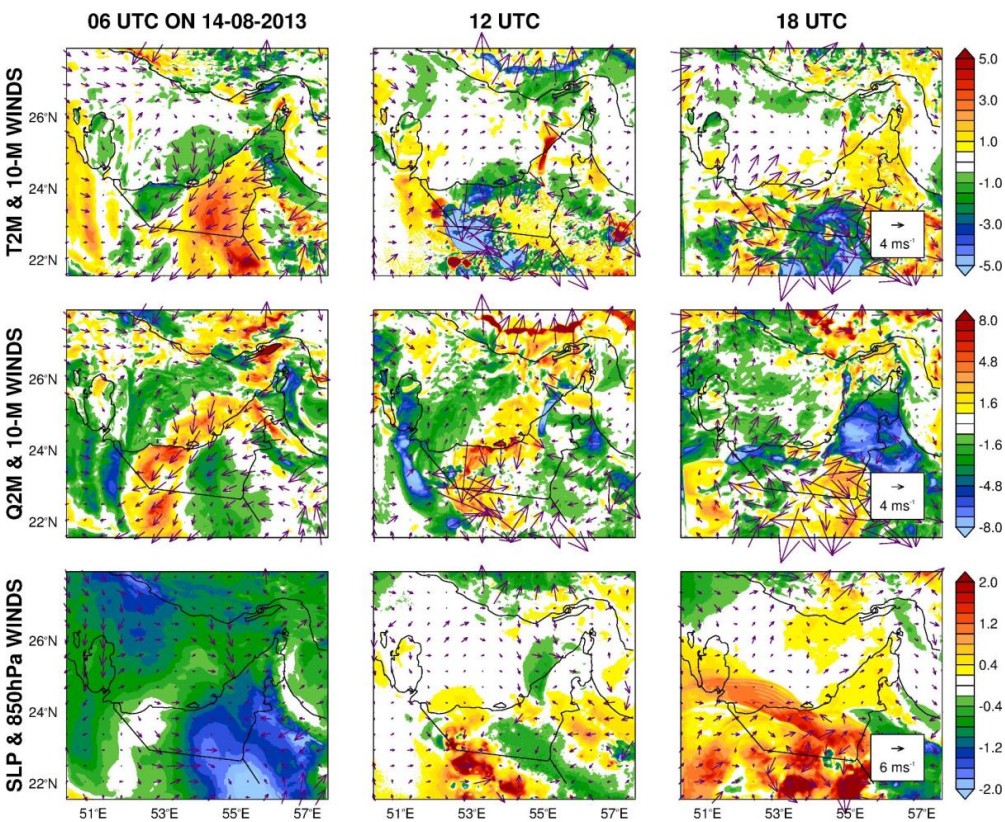

**Figure 9**: (a) 2-meter air temperature (K) and 10-meter wind vector (m s$^{-1}$) (top row), 2-meter water vapour mixing ratio (g kg$^{-1}$) and 10-meter wind vector (m s$^{-1}$) (middle row), and sea-level pressure (hPa) and 850 hPa wind vector (m s$^{-1}$) (bottom row) on 14 August 2013 at 06, 12 and 18 UTC for the the 2.5 km grid of the WRF-4 simulation. (b) and (c) are as (a) but showing the differences between runs WRF-5 and WRF-4 and WRF-6 and WRF-4, respectively.


### 4.2.3 Sensitivity of WRF forecasts to linear scaling of aerosol loading (WRF-6, 7, 10)


In this subsection, the impact of the aerosol loading on the WRF predictions of convection over the UAE
is analysed. Fig. 10 shows the upward and downward shortwave and longwave surface radiation fluxes
averaged over the whole of the UAE for all hours of day on 14 August 2013, for simulations WRF-6, WRF-
7 (as WRF-6 but scaling the aerosol loading by a factor of 5) and WRF-10 (as WRF-6 but multiplying the
amount of aerosols at all vertical levels by 10). The downward shortwave radiation flux at the surface



decreases in a roughly linear fashion as the aerosol loading is increased, with a drop of up to 180 $W\,m^{-2}$ for
WRF-7 and 360 $W\,m^{-2}$ for WRF-10 with respect to WRF-6, while the upward shortwave radiation flux is
cut by up to 40 $W\,m^{-2}$ and 81 $W\,m^{-2}$ for the same simulations, respectively. In a daily-averaged sense, and
with respect to WRF-6, the net shortwave radiation flux decreases by 46 $W\,m^{-2}$ in WRF-7 and 91 $W\,m^{-2}$ in
WRF-10. Assuming a linear scaling, for a doubling of the aerosol amount the change in the net shortwave
radiation flux would be about -18 $W\,m^{-2}$, in line with the values reported by Menut et al. (2019) for a study
over West Africa. On the other hand, the impact on the longwave radiation flux is much smaller, with hourly
changes in the net flux of up +62 $W\,m^{-2}$ and +129 $W\,m^{-2}$ for runs WRF-7 and WRF-10 with respect to WRF-
6, and daily-averaged values of +25 $W\,m^{-2}$ and +51 $W\,m^{-2}$, respectively. These changes are a factor of two
smaller than those estimated by Menut et al. (2019). This may be explained by the size distribution, to
which the longwave radiative forcing is known to be highly sensitive to (Adebiyi and Kok, 2020; Kok et
al., 2021), and hence the aerosol properties used in the model. As seen in Fig. 10, the downward longwave
radiation flux exhibits changes of less than ±10 $W\,m^{-2}$, as this field is mostly a function of the atmospheric
emissivity and cloud cover, both of which vary less than the surface temperature (Nelli et al., 2020a). The
upward longwave radiation flux, on the other hand, is lower for higher aerosol loadings as the surface
temperature drops, but the maximum reduction is still less than a factor of two to three smaller than the
decrease in the downward shortwave radiation flux. This is because the temperature does not vary much in
absolute values, as it is estimated from the surface energy budget in the model, with the different terms
adjusting to a changing downward shortwave radiation flux (Niu et al., 2011). As for the shortwave
radiation flux, the changes in the surface longwave radiation fluxes scale roughly linearly with the aerosol
loading, in line with Hansell et al. (2010).

The impact of the aerosol loading on the near-surface variables is summarized in Table 3, for the 35
NCM stations. The main difference between runs WRF-6, WRF-7 and WRF-10 is in the downward
shortwave radiation flux, with a bias of about +74 $W\,m^{-2}$, +9 $W\,m^{-2}$ and -46 $W\,m^{-2}$, respectively. The other
verification diagnostics, however, show very little changes between the simulations. The smaller bias for



WRF-7, which can be regarded as not-significant as $|\mu| << 0.5$, for which the observed and modelled
aerosol loadings (at least in the lower troposphere just above the surface) are comparable, Fig. 6c, highlights
the importance of properly capturing the aerosol amount for the simulation of the surface radiative fluxes.
The other variables given in Table 3 show much reduced relative changes between runs WRF-6, WRF-7
and WRF-10. In fact, the 2-meter temperature only decreases by about 0.5 K when the aerosol loading is
increased by a factor of 10, a similar variation reported by Menut et al. (2019) when the mineral dust
emissions are doubled. The surface temperature, on the other hand, is roughly 6 K colder in WRF-10
compared to WRF-6 (not shown). In the Noah-MP LSM, the air temperature is obtained from the surface
temperature, sensible heat flux and exchange coefficient for heat (Weston et al., 2018). The smaller change
in air temperature may be attributed to the decrease in the sensible heat flux, by about 32 W m$^{-2}$, leading to
comparable air temperature values. Besides, as the NCM stations are spread out over the UAE, Fig. 1c, and
as in some regions there is an increase in air temperature at certain times during the day due to drier
conditions (Fig. 10c), on an average sense the variation will be small. The increase in the aerosol loading
leads to warmer temperatures in the aerosol layer, with this being particularly evident at 12 UTC (Fig. A1a)
in particular below 700 hPa, where the concentration of aerosols is higher (Fig. 5d): the WRF temperature
biases increase from <0.5 K in WRF-6 to up to 3 K in WRF-10, and are accompanied by a drying of the
layer by up to 15 % (Fig. A1b).

As the aerosol loading is increased, the model-predicted precipitation decreases. This is true at the location
of the NCM stations (Table 3), and is easily seen in the accumulated precipitation maps, Figs. 8f, 8g and
8j, with a domain-wise reduction of roughly 1% and 16% in WRF-7 and WRF-10 with respect to WRF-6,
respectively. It can be explained by the aerosols' impact on the atmospheric circulation. A comparison of
Figs. 10c-d reveals that in WRF-10 the AHL is displaced to the east, with the associated circulation leading
to a deeper inland penetration of the moist Arabian Gulf air over western UAE and adjacent Saudi Arabia,
while the southeasterly winds ahead of it slow down the sea-breeze and lead to drier conditions over parts
of central and eastern UAE. Despite an aerosol loading that is 10 times higher, the drier environment here,





with differences in the water mixing ratio of more than 10 g kg$^{-1}$, allows for warmer air temperatures,
spreading into parts of the Gulf at 18 UTC. However, elsewhere it is colder in WRF-10 when compared to
WRF-6, in particular at 18 UTC. The reduced spatial extent and amount of the precipitation in WRF-10
arises from an eastward shift in the region of low-level wind convergence, into an area where the
atmosphere is drier. Figs. 10c-d highlight the importance of the aerosols' effects on the model-predicted
circulation (and consequently on the precipitation), which is more prominent for higher aerosol loadings, a
finding also reached e.g. by Lau et al. (2017). Besides the suppressed rainfall, there is also a delay in the
development of convective clouds as the aerosol loading is increased, as seen by comparing Figs. A2g-h
with A2k.

(a)

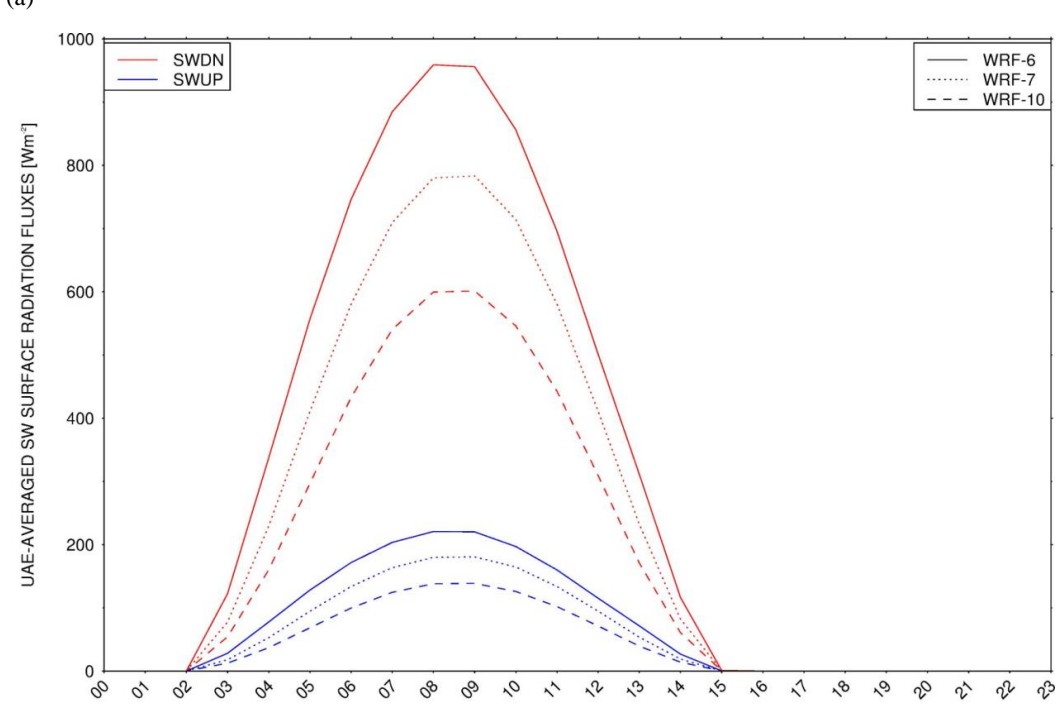

(b)




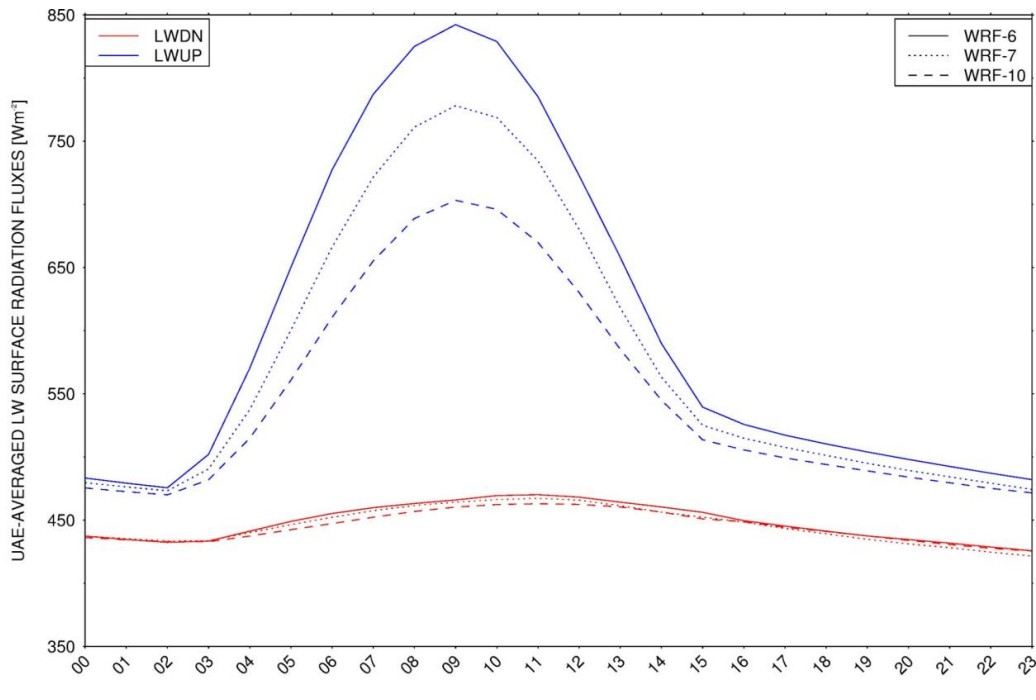

(c)





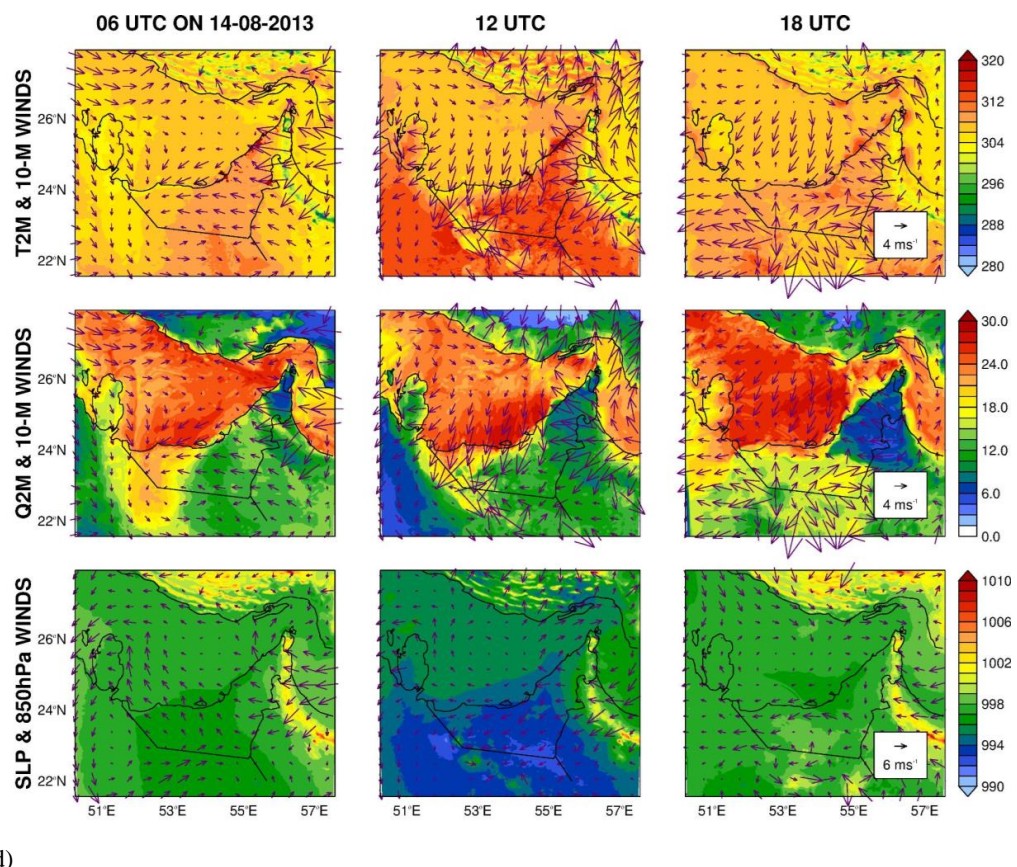

(d)



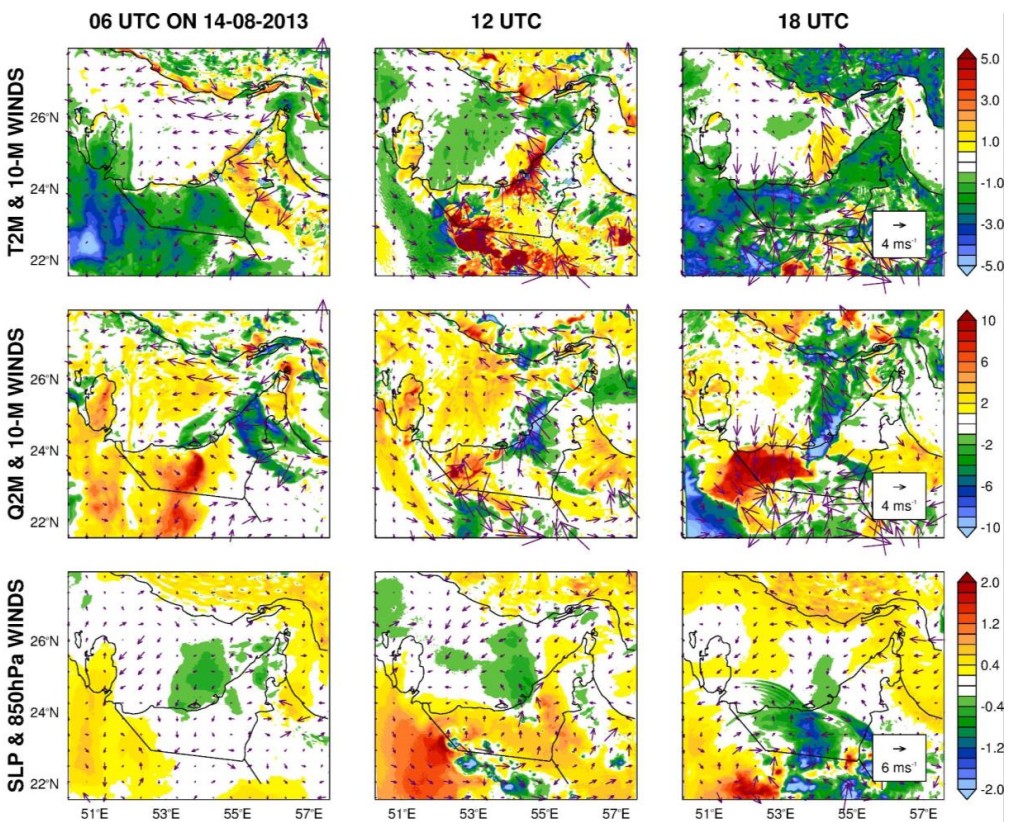

**Figure 10**: (a) UAE-averaged upward (red) and downward (blue) surface shortwave radiation flux (W m$^{-2}$) for 14 August 2013 for the simulations WRF-6 (solid line), WRF-7 (dotted line) and WRF-10 (dashed line). (b) is as (a) but for the longwave radiation fluxes. (c) 2-meter air temperature (K) and 10-meter wind vector (m s$^{-1}$) (top row), 2-meter water vapour mixing ratio (g kg$^{-1}$) and 10-meter wind vector (m s$^{-1}$) (middle row), and sea-level pressure (hPa) and 850 hPa wind vector (m s$^{-1}$) (bottom row) on 14 August 2013 at 06, 12 and 18 UTC for the for the 2.5 km grid of the WRF-6 run. (d) is as (c) but for the difference between simulations WRF-10 and WRF-6.


### 4.2.4 Sensitivity to aerosol properties (WRF-7 to WRF-9)


In section 4.2.3, the impact of the aerosol loading on the surface fluxes and atmospheric circulation was
investigated. Here, the focus will be on the aerosol properties, with the aerosol loading in all simulations
corresponding to that of the climatological distribution scaled by a factor of 5, which has been found to
give the best agreement with the observed PM10 values measured at the location of 16 weather stations
over the UAE, Fig. 6c. The results are summarized in Fig. 11 and in Table 3.





As stated in section 2.2, and due to the presence of carbonaceous particles, the urban aerosol model (WRF-
8) is more absorbing that the rural (default) model (WRF-7), while the maritime aerosol model (WRF-9) is
less absorbing as the larger particles are removed and some of the rural aerosols are replaced with sea salt.
The results in Figs. 11 a-b show that a change in the aerosol composition has a larger impact on the surface
radiation fluxes than a simple increase in the aerosol loading (cf. Figs. 10a-b). In particular, when the urban
aerosol model is used, the downward shortwave radiation flux is cut by up to 360 W m$^{-2}$ with a daily-
averaged reduction of around 114 W m$^{-2}$, a larger radiative effect than when the aerosol loading is multiplied
by a factor of 10. The important role played by the aerosol composition has also been highlighted e.g. by
Hodzic and Duvel (2018) for WRF simulations over Borneo. The reduction in the upward longwave
radiation flux, when compared to WRF-7, exceeds 100 W m$^{-2}$, and is a result of the much colder surface,
with the daily-averaged surface temperature dropping by about 7 K and the air temperature by 0.8 K (Table
3). The radiation absorbed by the aerosols during the day is emitted at night, and in the urban aerosol model,
the aerosols are so absorbing that the surface downward longwave radiation flux in WRF-8 is up to 12 W
m$^{-2}$ higher than in WRF-7 at night, Fig. 11b. The impact of changing aerosol properties on the temperature
and RH vertical profiles is given in Fig. A1. The most noteworthy difference between simulations WRF-7
and WRF-8 is the heating around 700-750 hPa and the cooling below 800 hPa in simulation WRF-8 at 12
UTC, of magnitudes up to +1.5 K and -3.5 K, respectively. As the urban aerosols are more absorbing, and
most are below 700 hPa at this time (Fig. 5d), there is a strong heating at the top of the layer and a cooling
at lower levels as the vast majority of the incoming solar radiation is absorbed. This is in contrast when the
aerosol loading is increased, where the most pronounced warming occurs in the lowest part of the layer.

The impact of making the aerosols more absorbing on the atmospheric circulation is presented in Figs. 11c-
d. When carbonaceous aerosols are added, the AHL is weaker (note the anticyclonic circulation in the 10-
meter winds at 06 UTC and to a lesser extent at 12 UTC) and broader (note the negative sea-level pressure
anomalies over the Arabian Gulf and Oman) in WRF-8 when compared to WRF-7. This is consistent with
the referred pronounced reduction in the downward shortwave radiation flux and resulting colder surface





and air temperatures (Table 3). As the land temperatures become more comparable to the sea surface skin
temperatures over the Gulf, the sea-level pressure minimum extends into adjacent areas, which allows the
AHL to expand. As a result of the modifications to the AHL, the excessive moistening over western UAE
is reduced, and increased over eastern and southeastern parts of the country. The interaction between the
associated cyclonic circulation and the sea-breeze from the Sea of Oman and Arabian Gulf leads to a region
of low-level wind convergence here, where, and also due to a more moist environment, the model predicts
precipitation (Fig. 8h). WRF-8 is the wettest simulation over the UAE, with roughly 35% of the observed
precipitation at the location of the NCM stations captured by WRF (Table 3). However, a comparison of
Figs. 8h-i reveals that the rainfall falls from shallower clouds, with deep convection virtually absent in this
simulation. The weakening of the AHL also brings it closer to that given by ERA-5, Fig. 6b.

When the maritime aerosol model is used, on the other hand, there is a small increase in the downward
shortwave radiation flux by up to $75\,\mathrm{W\,m^{-2}}$ (by ~$22\,\mathrm{W\,m^{-2}}$ on a daily-averaged scale), with a roughly 1 K
increase in the surface temperature at the location of the NCM stations (not shown). The AHL is slightly
weaker and smaller in size in this run, Fig. 11d, albeit the changes in sea-level pressure are mostly within
1 hPa, whereas in WRF-8 in some regions they exceed 2 hPa. As a result, the precipitation and the clouds
shift southwards with respect to that in WRF-7, Figs. 8g and 8i and Figs. A2h and A2j, with less rainfall
accumulated at the location of the NCM stations (Table 3).






(a)

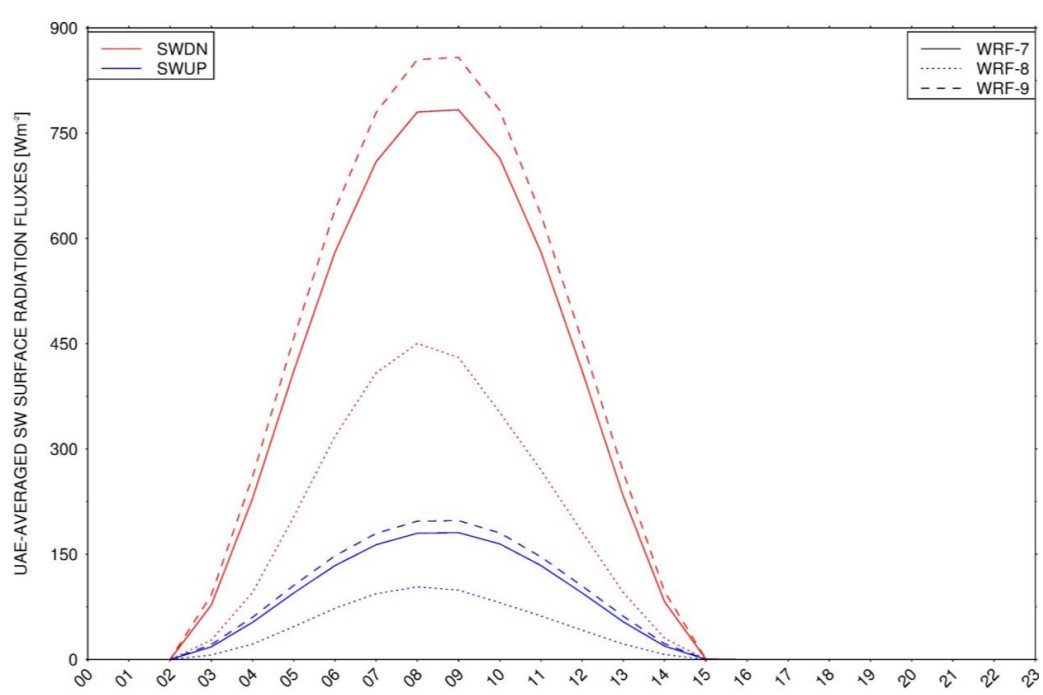

(b)





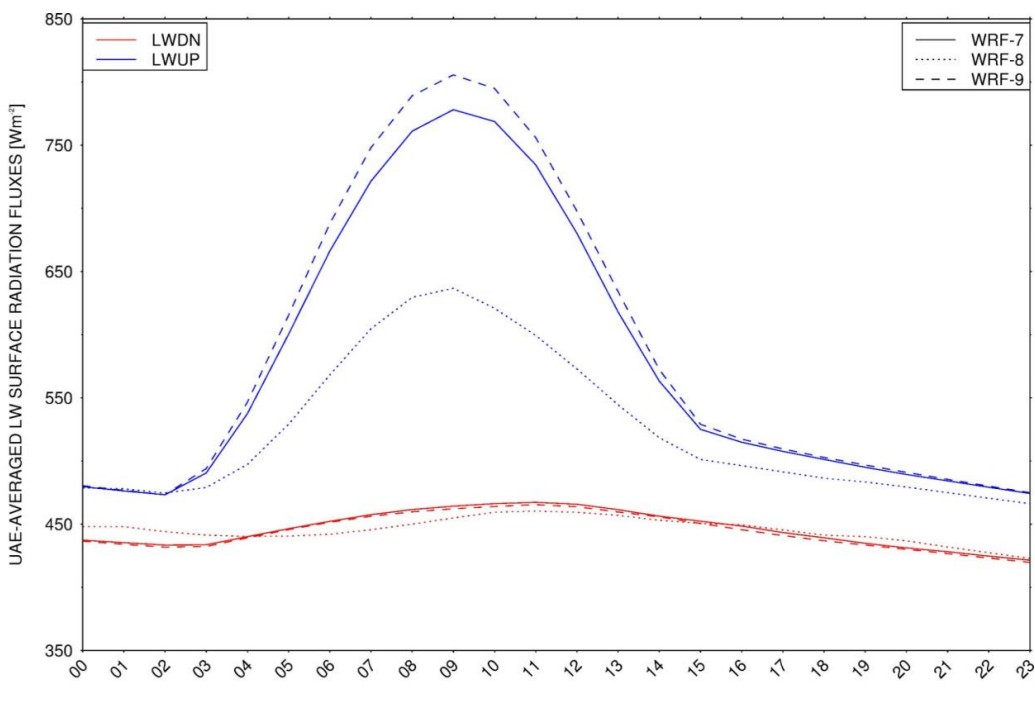

(c)



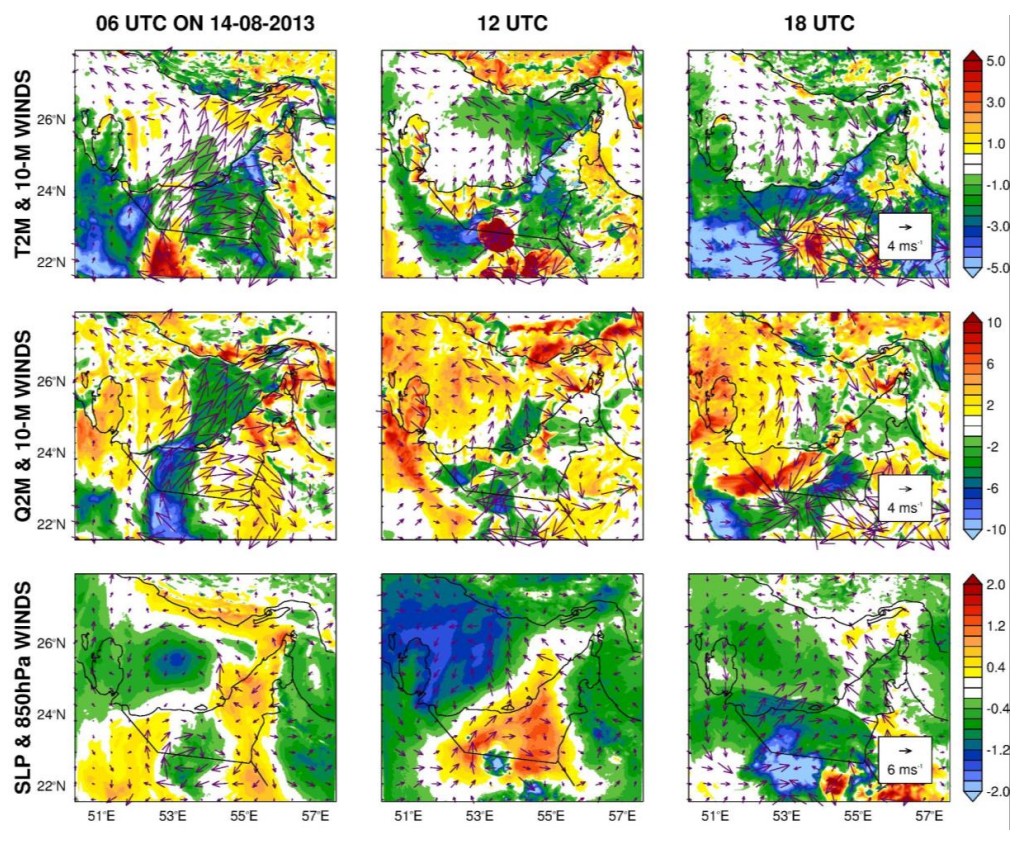

(d)

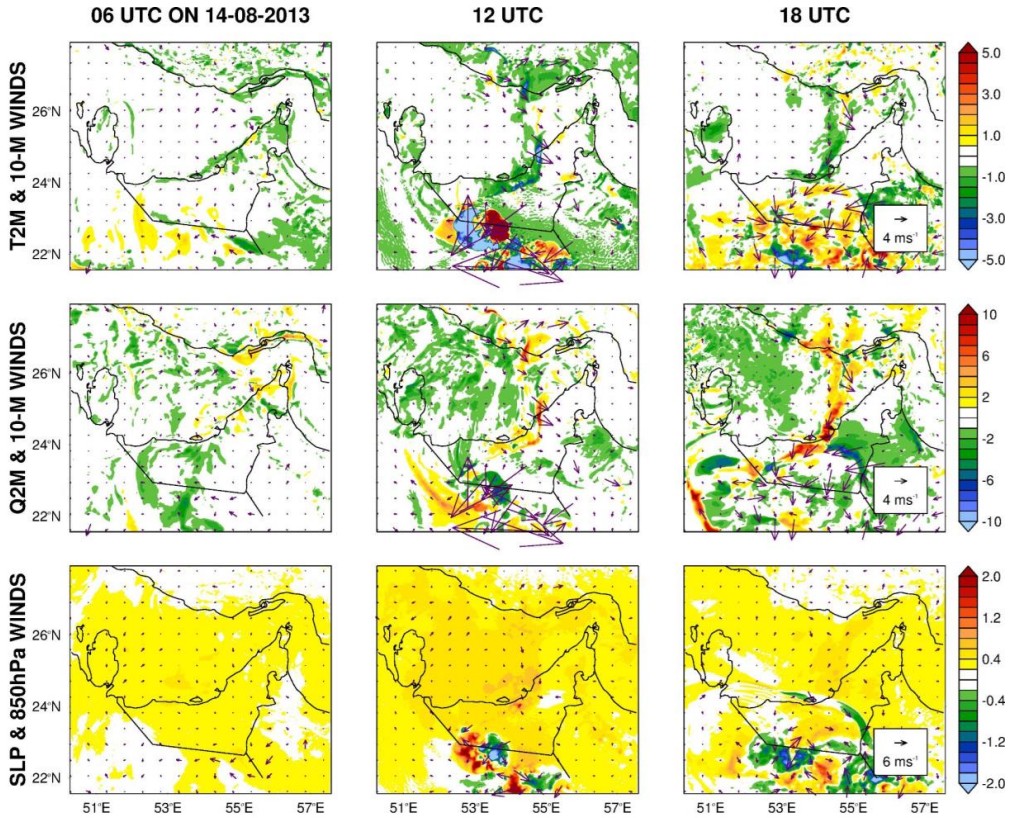

**Figure 11:** (a) UAE-averaged upward (red) and downward (blue) surface shortwave radiation flux (W m$^{-2}$) for 14 August 2013 for the simulations WRF-7 (solid line), WRF-8 (dotted line) and WRF-9 (dashed line). (b) is as (a) but for the longwave radiation fluxes. (c) differences in the 2-meter air temperature (K) and 10-meter wind vector (m s$^{-1}$) (top row), 2-meter water vapour mixing ratio (g kg$^{-1}$) and 10-meter wind vector (m s$^{-1}$) (middle row), and sea-level pressure (hPa) and 850 hPa wind vector (m s$^{-1}$) (bottom row) on 14 August 2013 at 06, 12 and 18 UTC for the for the 2.5 km grid between the WRF-8 and WRF-7 runs. (d) is as (c) but for the difference between simulations WRF-9 and WRF-7.






## 5. Discussion and Conclusions

In this manuscript, the Weather Research and Forecasting (WRF) model is used to investigate the role played by the aerosol loading and properties in a dusty summertime convective event in the United Arab Emirates (UAE), which took place on 14 August 2013. WRF is run in a three-nest 22.5 - 7.5 - 2.5 km configuration, and its predictions are assessed against ERA-5 reanalysis data and in-situ meteorological measurements at the location of 35 weather stations over the UAE. This convective event was triggered by the low-level convergence of the cyclonic circulation associated with the Arabian Heat Low (AHL), located over western UAE, and the sea-breeze from the Arabian Gulf and Sea of Oman. Hence, a correct simulation of the position and strength of the AHL is needed for the model to successfully capture the observed convective clouds. The 14 August 2013 was also a rather dusty day in the UAE, with Aerosol Optical Depths (AODs) in excess of two. An analysis of reanalysis data revealed that two factors played a role in the dust lifting activities on this day: (i) cold pools and downbursts, which occurred in association with the convective activity in the local afternoon and evening hours, and (ii) strong near-surface winds earlier in the day resulting from the interaction between the AHL and sea-land breeze circulations, with the placement of the ITD along the UAE favouring dust lifting. The dusty air in the afternoon was mostly trapped over western UAE by the AHL circulation, with a gradual decrease in the AODs going into the evening and nighttime hours arising from the advection of clearer air into the area as well as the clearing of the air due to the occurrence of precipitation.

The main findings of this work are as follows:

➢ Two aerosol distributions are considered in this study: an idealised distribution, set up for the continental United States, and a climatological profile, based on a 7-year output of a general circulation model. Even though the aerosol loading in the latter is roughly an order of magnitude larger, in both it is smaller than that observed at individual weather stations. The best agreement with that observed is obtained when the climatological values are multiplied by a factor of 5, in line



with the dustier atmosphere during this event. The skill scores in this run, and with respect to near-surface weather variables measured at the location of the 35 weather stations, are generally the highest out of the 10 simulations performed here;

➢ For the simulations with the idealised and climatological aerosol distributions, when the ARI effects are switched on, the daily-averaged surface downward shortwave radiation flux in the is reduced by 3 W m$^{-2}$ and 20 W m$^{-2}$, respectively, leading to changes in the surface temperature within 1 K and in the air temperature within 0.5 K. On the other hand, there is a heating of the atmosphere in the aerosol layer by up to 0.5 K. Activating the ARI effects when the climatological aerosol loading is used leads to a roughly 47% increase in the domain-wide precipitation, as the convective cells are more active, and the stronger updraft increases the fraction of activated aerosols.

➢ WRF has a pronounced cold bias over the UAE, with the daily-averaged air temperature being about 2.5 K colder at the location of individual weather stations compared to the observed values. As changes to the model physics (e.g. Weston et al., 2018) and boundary conditions (e.g. Temimi et al., 2020) do not alleviate the problem, employing interior nudging in the outermost and two outermost grids was considered. While the skill scores of the innermost nest improved in particular when interior nudging was applied to the two outermost grids, in line with Wootten et al. (2016), the cold bias in the 2.5 km grid persisted. This is because a change in the atmospheric circulation, in particular in the position of the AHL, leads to increased precipitation over the UAE and locally colder temperatures, which offset the higher temperatures that arise from more accurate boundary conditions. Nudging also leads to a margin improvement in the model temperature and RH profiles with respect to those estimated from radiosondes launched at Abu Dhabi's International Airport, with the respective biases decreasing by up to 1 K and 10%, respectively;

➢ The downward and upward shortwave and the upward longwave radiation fluxes are found to decrease linearly as the aerosol loading is increased, with a 10-fold increase in the amount of



aerosols leading to a daily-averaged drop of the surface net shortwave flux of about 91 W m$^{-2}$ and
an increase in the net longwave radiation flux by roughly 51 W m$^{-2}$. The surface temperature
decreases by about 6 K and the air temperature by 0.5 K, with a warming of up to 3 K in the aerosol
layer. As the aerosol loading goes up, the AHL shifts eastwards, with the low-level wind
convergence taking place in a drier region, resulting in lower precipitation amounts falling in a
more spatially confined area. In addition, the onset of convection is also delayed;
➢ When 20% of the aerosols are replaced with more absorbing (carbonaceous) particles, the roughly
87 W m$^{-2}$ decrease in the surface net shortwave radiation flux is comparable to the drop when the
aerosol loading is augmented by a factor of 10. This stresses that the aerosol composition plays a
role as important as its amount on the surface radiative fluxes, in line with other studies such as
Hodzic and Duvel (2018), at least for the range of values considered here. The atmospheric
response, on the other hand, is very different, with a weaker and broader AHL allowing for a deeper
sea-breeze penetration and increased amount of rainfall over the UAE. As during daytime, the
aerosol concentration is roughly uniform below 700 hPa due to strong vertical mixing, there is a
warming up to 1.5 K in the upper aerosol layer, and a cooling of up to 3.5 K at the lowest part of
the layer, when the aerosols are made more absorbing. When the aerosol loading is increased, on
the other hand, the warming has a higher magnitude in the lowest part of the layer. The sensitivity
to the maritime aerosol model, for which 20% of the rural aerosols are replaced by sea-salt and the
larger particles removed, on the other hand, is much reduced.

Despite the higher spatial resolution at which the WRF was run, and the use of an optimized set up for
summertime convective events in the region with an improved representation of the lower boundary
conditions, the model still failed to capture the observed convective clouds and associated precipitation.
What is more, the representation of aerosol-radiation and aerosol-cloud interactions in the model still need
to be further refined, in particular with respect to the aerosol optical properties and size distribution. This



can be achieved through additional studies that combine both in-situ measurements (such as aerosol
concentration profiles from aircraft measurements, i.e., Wehbe et al., 2021) and numerical modelling. In
any case, the experiments conducted in this study stress that, while it is important to capture the observed
aerosol loading for a correct simulation of the surface radiative fluxes, changes in both the amount and
composition of the aerosols will have an important impact on the atmospheric circulation, convection and
precipitation. For the case of this convective event, where the model-predicted rainfall is very sensitive to
the position and strength of the thermal low, such an effect on the circulation has a rather large impact on
the predicted convective regions. An extension of this work would be to investigate whether similar findings
are reached for summertime convective events that occur on the eastern side of the UAE, for which the
AHL plays a reduced role in the triggering of the convective clouds (e.g., Francis et al., 2021). This will be
left to a future study.



**Acknowledgements**

This work is supported by the National Center of Meteorology (NCM), Abu Dhabi, United Arab Emirates (UAE), under the UAE Research Program for Rain Enhancement Science (UAEREP). The authors thank the NCM for providing the weather station and air quality observations over the UAE, under an agreement with clauses for non-disclosure of data. Access to these data is restricted and readers should request them through contacting research@ncms.ae. We would also like to thank the UAE NCM for kindly providing radiosonde data at Abu Dhabi's International Airport through the National Oceanic and Atmospheric Administration Integrated Global Radiosonde Archive's website. The authors wish to acknowledge the major contribution of Khalifa University's high-performance computing and research computing facilities to the results of this research. We are very grateful to the Environmental Agency - Abu Dhabi for providing hourly air quality measurements over Abu Dhabi. Readers who wish to access this data are requested to e-mail customerhappiness@ead.gov.ae.



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





**Appendix**



(a)

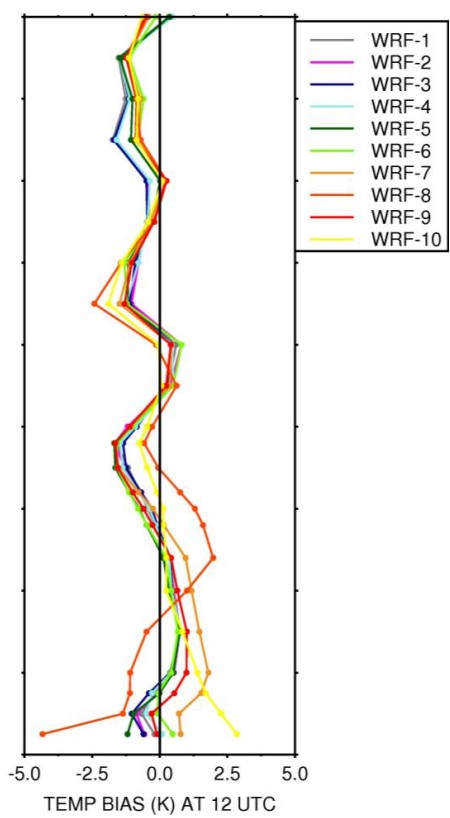

(b)

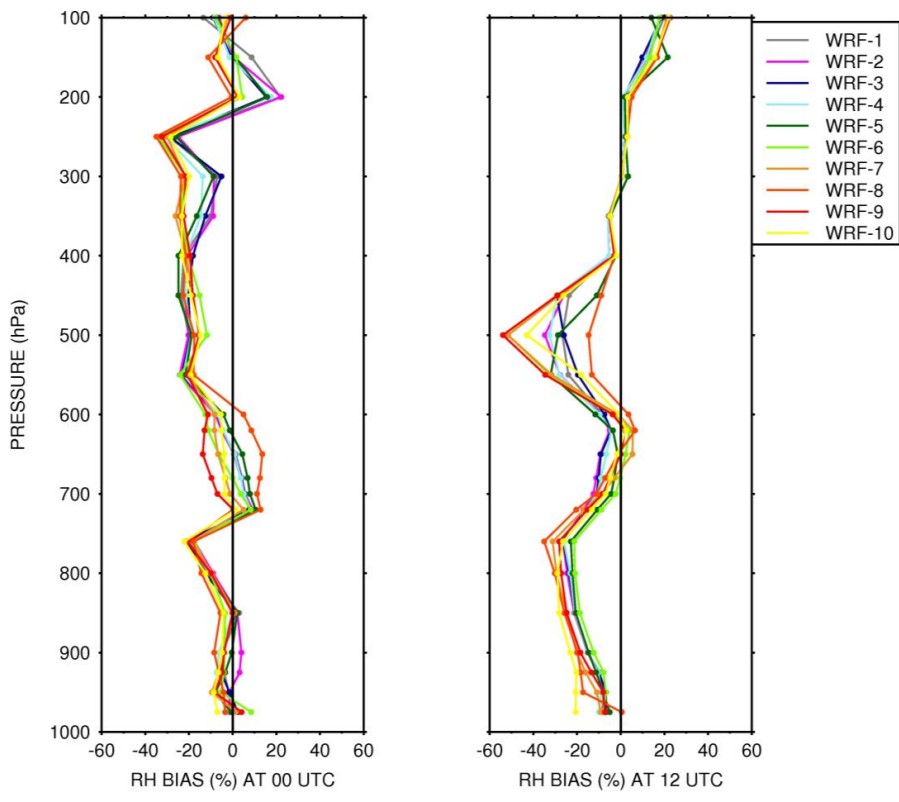

**Figure A1**: Bias of the WRF-1 to WRF-10 vertical profiles of (a) temperature (K) and (b) relative humidity (%) with respect to the radiosonde profiles launched at Abu Dhabi's airport on 14 August 2013 at 00 and 12 UTC. The solid black vertical line in all panels gives the optimal score (i.e. zero bias).


(a)

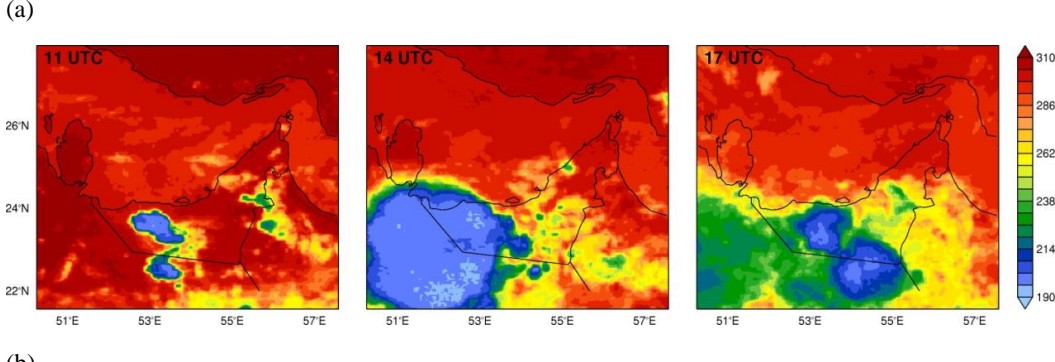

(b)









(g)

(h)

(i)

(j)

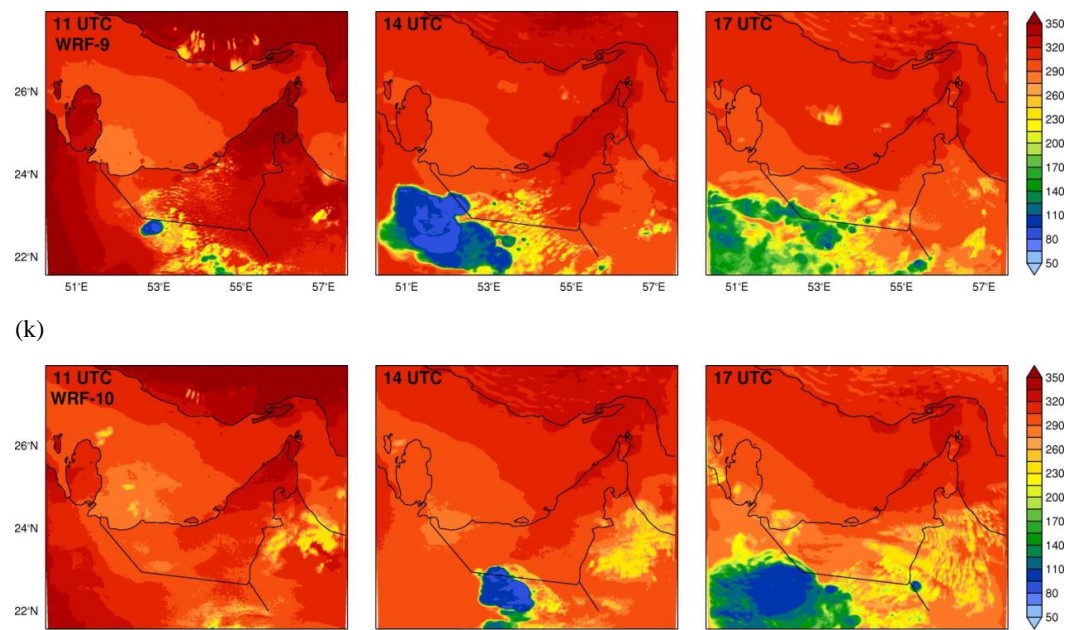

(k)

**Figure A2**: (a) IRBT (K) on 14 August 2013 at 11, 14 and 17 UTC. (b)-(k) Outgoing Longwave Radiation (OLR; W m$^{-2}$) on the same day and at the same times for simulations WRF-1 to WRF-10, respectively.


(a)

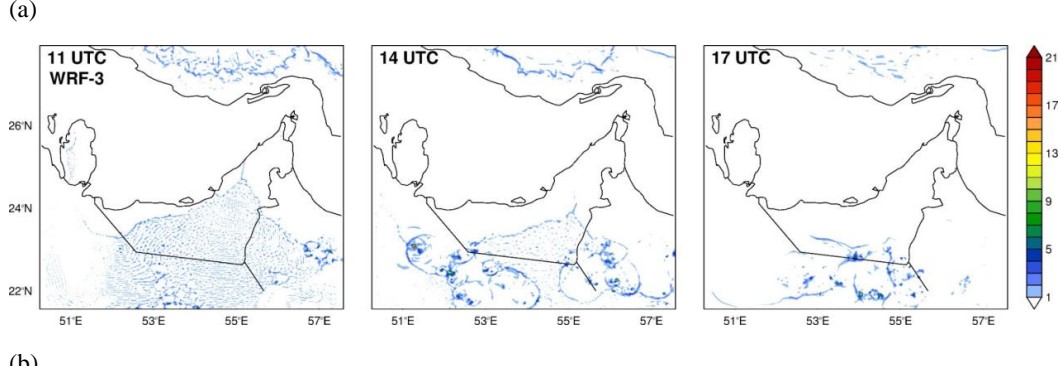

(b)



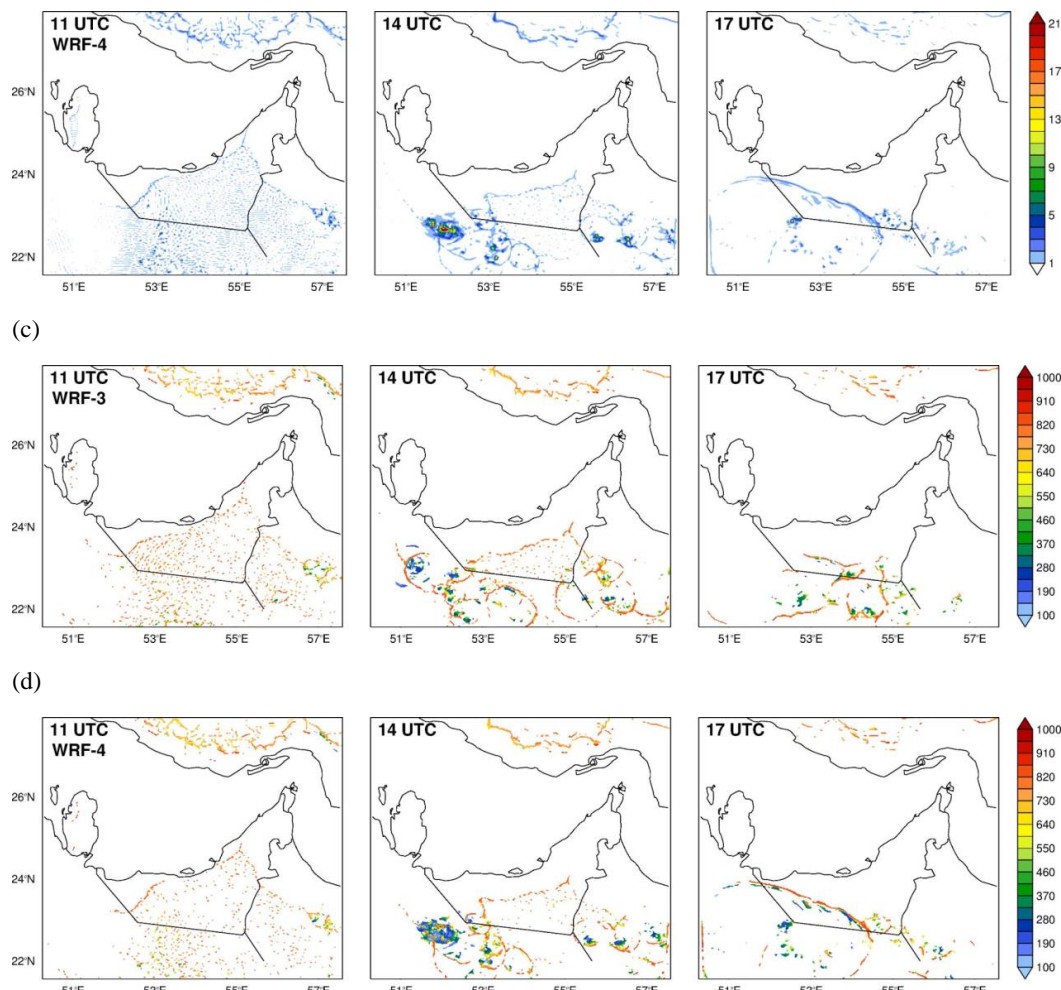

**Figure A3**: (a) Maximum vertical velocity (m s⁻¹) in the column of the 2.5 km WRF grid for simulation WRF-3. (b) is as (a) but for WRF-4. (c) Pressure level (hPa) at which the maximum vertical velocity is observed in WRF-3. (d) is as (c) but for WRF-4.
