# Peer review of "Convection-Aerosol Interactions in the United Arab Emirates: A Sensitivity Study"

_Atmospheric Chemistry and Physics, 2021_

## Referee Comment (RC1)

*Comments on the manuscript entitled "Convection-Aerosol Interactions in the United Arab Emirates: A Sensitivity Study" by Ricardo Fonseca, et al.*

**Recommendation:** Major Revisions

**General comments:**

This manuscript 'Convection-Aerosol Interactions in the United Arab Emirates: A Sensitivity Study' mainly investigate the impacts of aerosol loading and properties on the atmospheric circulation, convective activity, surface/air temperature, and local precipitation by Weather Research and Forecasting (model) in UAE on 14 August 2013. The authors carried out ten different scenarios for WRF simulations and compared the different results of circulation, radiative effect, convective, and rainfall. In general, the paper presents in a logical way, but the English writing need to be greatly improved. Some interesting results of this manuscript will be helpful to understanding the interactions between the convection and aerosol. I therefore recommend publication of this paper in **Atmospheric Chemistry and Physics** after major revisions. My comments are listed as follows.

**Major Comments:**

1. Compared with the previous published papers, what are the main innovations of this manuscript? Please elucidate clearly in the context.

   Many conclusions of this manuscript are consistent with the previous publications. For instance,

   **(Page 1, Abstract, lines 13-15)** 'The convection on 14 August 2013 was triggered by the low-level convergence of the circulation associated with the Arabian Heat Low (AHL) and the daytime sea-breeze circulation.' **This conclusion is the same as the previous publications in (Page 3, 1. Introduction, Lines 113-116.)** 'As discussed in Schwitalla et al. (2020) and Branch et al. (2020), it is normally triggered by the convergence of the low-level circulation associated

with the Arabian Heat Low (AHL; Fonseca et al., 2021), the sea-breeze circulation from the Arabian Gulf and Sea of Oman, and the upslope flows on the mountains.' (**Page 6, 1. Introduction, Lines 123-124.**) 'Here, they are commonly triggered by the low-level convergence of the AHL and sea-breeze circulations (Steinhoff et al., 2018)'.

(**Page 2, Abstract, lines 31-32 and the Conclusions**) 'The surface downward and upward shortwave and upward longwave radiation fluxes are found to scale linearly with the aerosol loading, ….'   **This conclusion is consistent with** (**Page 4, 1. Introduction, Lines 80-84.**) 'Liu et al. (2020) used the WRF model with Chemistry (WRF-Chem; Grell et al., 2005) to investigate the effects of biomass burning aerosol on radiation, clouds and precipitation in the Amazon basin. The authors found that ACI effects prevail at lower emission rates and low values of aerosol optical depth (AOD), while the ARI plays the largest role at high emission rates and high AODs.'

2. (**Page 11, 2.2 WRF Experiments and the whole context**): The authors implemented 10 different scenarios for WRF simulations based on two aerosol distributions (an idealized aerosol distribution profile and a climatological profile) and compared the different impacts of aerosol loading and optical properties on the atmospheric circulation, radiative effect, convective, and rainfall. The authors carried out a lot of simulations for sensitivity experiments and acquired many conclusions, **but it is not clear for the readers, which conclusion is important and which one is close to the observed results for this manuscript**.

  For instance, (1) **Page 56, 5. Discussion and Conclusions, Lines 855-856**, 'The best agreement with that observed is obtained when the climatological values multiplied by a factor of 5, in line with the dustier atmosphere during this event'. (2) **Pages 57-58, Lines 879-882**, 'The downward and upward shortwave and the upward longwave radiation fluxes are found to decrease linearly as the as aerosol loading is increased, with a 10-fold increase in the amount of aerosols leading to a

daily-averaged drop of the surface net shortwave flux of about 91 Wm$^{-2}$, and …….'.

(3) **Page 58, Lines 887-889**, 'When 20% of the aerosols are replaced with more absorbing (carbonaceous) particles, the roughly 87 Wm$^{-2}$ decrease in the surface net shortwave radiation flux…when the aerosol loading is augmented by a factor of 10'.

(4) **Page 58, Lines 897-899**, 'The sensitivity to the maritime aerosol model, for which 20% of the rural aerosols are replaced by sea-salt and the larger particles removed, on the other hand, is much reduced.'

Whether the changes of aerosol loading and optical properties in the WRF sensitivity simulations could reflect the true observations or not?

In this manuscript, the authors indicated that 'The 14 August 201 was also a rather dusty day in the UAE, with Aerosol Optical Depths (AODs) in excess of two', and **I suggest the authors should implement the sensitivity of the potential effects of dust aerosols' loadings and optical properties on the circulation, convection, radiative forcing, and precipitation**.

3. In WRF simulations of this manuscript, how to consider the potential influences of environmental field (e.g. wind speed field, air humidity field), and vertical convection on the ARI, ACI, circulation, convection activity, and precipitation, etc?

4. The English written of this whole manuscript need to be greatly improved.

**Minor comments:**

**1. Page 3, lines 58-60:** 'Dust has been shown to have an important impact on the climate system, in particular on the atmosphere (e.g. Min et al., 2014; Liu et al., 2019; Francis et al., 2020), ocean (e.g. Evan et al., 2012) and cryosphere (e.g. Francis et al., 2018) dynamics.'

⇒ Please delete all the 'e.g.' in the cited literatures, and modify the other places in the context.

**2.** When talking about the direct and semi-direct radiative effects of aerosols, the authors could cite other references,

[1] Li Z., Y. Wang, J. Guo, et al. 2019: East Asian study of tropospheric aerosols and their impact on regional clouds, precipitation, and climate (EAST-AIR(CPC)). *Journal of Geophysical Research: Atmospheres*. 124 (23), 13026-13054. DOI: 10.1029/2019JD030758.

[2] Wang W., J. Huang, P. Minnis, et al. 2010: Dusty cloud properties and radiative forcing over dust source and downwind regions derived from A-Train data during the Pacific Dust Experiment. *Journal of Geophysical Research: Atmospheres*. 115 . DOI:10.1029/2010JD014109.

---

## Author Comment (AC1)

**ACP-2021-597: Convection-Aerosol Interactions in the United Arab Emirates: A Sensitivity Study**

**By Fonseca et al. (2021)**

**Reply to reviewer #1's comments:**

This manuscript 'Convection-Aerosol Interactions in the United Arab Emirates: A Sensitivity Study' mainly investigate the impacts of aerosol loading and properties on the atmospheric circulation, convective activity, surface/air temperature, and local precipitation by Weather Research and Forecasting (model) in UAE on 14 August 2013. The authors carried out ten different scenarios for WRF simulations and compared the different results of circulation, radiative effect, convective, and rainfall.

In general, the paper presents in a logical way, but the English writing need to be greatly improved. Some interesting results of this manuscript will be helpful to understanding the interactions between the convection and aerosol. I therefore recommend publication of this paper in Atmospheric Chemistry and Physics after major revisions. My comments are listed as follows.

**REPLY:** The authors would like to thank the reviewer for his/her valuable comments/suggestions, which helped to improve the quality of the manuscript. Following his/her feedback, we have rephrased poorly written and potentially confusing sentences and put more emphasis on the main findings of the study. Below we address the reviewer's queries one by one, highlighting in the text where changes, if any, were made.

**Major Comments:**

1. Compared with the previous published papers, what are the main innovations of this manuscript? Please elucidate clearly in the context.

Many conclusions of this manuscript are consistent with the previous publications. For instance,

- (**Page 1, Abstract, lines 13-15**) 'The convection on 14 August 2013 was triggered by the low-level convergence of the circulation associated with the Arabian Heat Low (AHL) and the daytime sea-breeze circulation.' **This conclusion is the same as the previous publications in** (**Page 3, 1. Introduction, Lines 113-116.**) 'As discussed in Schwitalla et al. (2020) and Branch et al. (2020), it is normally triggered by the convergence of the low-level circulation associated with the Arabian Heat Low (AHL; Fonseca et al., 2021), the sea-breeze circulation from the Arabian Gulf and Sea of Oman, and the upslope flows on the mountains.'
- (**Page 6, 1. Introduction, Lines 123-124.**) 'Here, they are commonly triggered by the low-level convergence of the AHL and sea-breeze circulations (Steinhoff et al., 2018)'.
- (**Page 2, Abstract, lines 31-32 and the Conclusions**) 'The surface downward and upward shortwave and upward longwave radiation fluxes are found to scale linearly with the aerosol loading, ....' **This conclusion is consistent with** (**Page 4, 1. Introduction, Lines 80-84.**) 'Liu et al. (2020) used the WRF model with Chemistry (WRF-Chem; Grell et al., 2005) to investigate the effects of biomass burning aerosol on radiation, clouds and precipitation in the Amazon basin. The authors found that ACI effects prevail at lower emission rates and low values of aerosol optical depth (AOD), while the ARI plays the largest role at high emission rates and high AODs.'

**REPLY:** We thank the reviewer for his/her comment.

The statements in the first two bullet points above are related to the triggering mechanisms of the convective event considered in this study. As noted in the Introduction (lines 112-134), there are two main types of summertime convective events in the United Arab Emirates: on the eastern side around the Al Hajar mountains, for which the cloud development is aided by the topographic circulation (Branch et al., 2020, Francis et al., 2020), and on the western side, where clouds typically arise from the low-level convergence between the Arabian Heat Low circulation (Fonseca et al., 2021) and the sea-breeze from the Arabian Gulf (Steinhoff et al., 2018). The 14 August 2013 event falls into the latter category, and therefore the triggering mechanism is expected to be consistent with that of previous studies of similar convective events. We would also like to stress that the sentences the reviewer refers to here are mostly in the Introduction, where a literature review is normally given, while in the remaining of the paper, and in particular in the Discussion and Conclusions section, the focus is on the novel results of the work.

Regarding the sentences in the third bullet point, Liu et al. (2020) did not reach the conclusion that the surface downward/upward shortwave and upward longwave radiation fluxes scale linearly with the aerosol loading. Instead, the authors focused on the aerosol-cloud and aerosol-radiation interaction effects on precipitation, which we also discuss in our study. As a result, that particular conclusion is novel and worth being reported in the Abstract.

Given the reviewer's comment, and in particular in the Discussion and Conclusions section, we have put greater emphasis on the novel findings of the study (lines 790-835) and now clearly highlight the take-home messages for potential future readers of this work (lines 837-853).

2. (**Page 11, 2.2 WRF Experiments and the whole context**): The authors implemented 10 different scenarios for WRF simulations based on two aerosol distributions (an idealized aerosol distribution profile and a climatological profile) and compared the different impacts of aerosol loading and optical properties on the atmospheric circulation, radiative effect, convective, and rainfall. The authors carried out a lot of simulations for sensitivity experiments and acquired many conclusions, **but it is not clear for the readers, which conclusion is important and which one is close to the observed results for this manuscript**.

For instance, (1) **Page 56, 5. Discussion and Conclusions, Lines 855-856**, 'The best agreement with that observed is obtained when the climatological values multiplied by a factor of 5, in line with the dustier atmosphere during this event'. (2) **Pages 57-58, Lines 879-882**, 'The downward and upward shortwave and the upward longwave radiation fluxes are found to decrease linearly as the as aerosol loading is increased, with a 10-fold increase in the amount of aerosols leading to a daily-averaged drop of the surface net shortwave flux of about 91 $Wm^{-2}$, and …….'. (3) **Page 58, Lines 887-889**, 'When 20% of the aerosols are replaced with more absorbing (carbonaceous) particles, the roughly 87 $Wm^{-2}$ decrease in the surface net shortwave radiation flux…when the aerosol loading is augmented by a factor of 10'. (4) **Page 58, Lines 897-899**, 'The sensitivity to the maritime aerosol model, for which 20% of the rural aerosols are replaced by sea-salt and the larger particles removed, on the other hand, is much reduced.'

**REPLY:** We fully agree with the reviewer we have to emphasize the main findings of the study. As stated in the reply to his/her previous comment, we have now made it abundantly clear in the Discussion and Conclusions section our main findings and take-home messages for readers of this work (lines 790-853).

Regarding a comparison with the observational measurements, no simulation clearly outperformed another. In fact, Table S1 shows that, by and large, WRF's cold and dry biases are present in all model runs, and readers interested in running WRF over hyper arid regions need to be aware of this bias. Having said that, we are in a position to issue recommendations for users interested in running the WRF model for such

convective events in hyper-arid regions located next to major dust sources like the United Arab Emirates (UAE):

➢ When accounting for the observed aerosol loading, using a climatology-based distribution is preferable to an idealized distribution as it can improve the representation of deep convection, as evidenced by the increased precipitation generated by the model and the colder cloud tops, in particular when the aerosol-radiation interaction (ARI) effects are switched on. The vertical profiles of variables like temperature are also better simulated;

➢ Even in short-term (2-day) simulations, the fields in the interior of the WRF nests can be substantially different from those in the input (in this case reanalysis) dataset. Employing nudging in the outer nests (in this case in the first two model grids) is preferable to only applying it in the outer nest or not doing it altogether, as it helps to at least partially correct some of the WRF biases;

➢ It is vital to accurately represent the properties of the observed aerosols in the model, more so than the amount, provided the order of magnitude is in line with that observed. If the aerosols are more absorbing, the heating in the aerosol layer will peak closer to its top instead of in the bottom half, which has implications for the dynamics and convection in particular if the aerosol layer is deep and/or multiple layers are present.

We have stated this in the text (lines 837-853) and would like to thank the reviewer again for raising this issue.

Whether the changes of aerosol loading and optical properties in the WRF sensitivity simulations could reflect the true observations or not?

**REPLY:** We thank the reviewer for raising this point. In Fig. 6c we compare the model-predicted aerosol optical depth (AOD) with that given by the MERRA-2 reanalysis dataset. This particular dataset explicitly accounts for aerosols and their interactions with the climate system, and is found to perform well in this region (Roshan et al., 2019; Ukhov et al., 2020). While we can get the correct order of magnitude when scaling the climatological aerosol loading by a factor of five, the diurnal trend in the reanalysis dataset is not simulated by the Weather Research and Forecasting (WRF) model. We speculate on why this may be the case (lines 504-518). Due to the extensive cloud cover on 14 August 2013 (Figs. 2a-c), AOD estimates from ground and satellite assets exhibit gaps and missing data and hence cannot be used to directly evaluate the WRF predictions. What is more, we do not have aircraft measurements of aerosol loading at different heights to assess the vertical distribution nor information regarding its optical properties. We understand this is a limitation of the study and have noted it in the text (lines 504-510). We believe a comprehensive assessment of the simulated aerosol loading and properties would require additional observational data that is not available for us. This has also been highlighted in the Discussion and Conclusions section (lines 858-863).

In this manuscript, the authors indicated that 'The 14 August 201 was also a rather dusty day in the UAE, with Aerosol Optical Depths (AODs) in excess of two', and **I suggest the authors should implement the sensitivity of the potential effects of dust aerosols' loadings and optical properties on the circulation, convection, radiative forcing, and precipitation**.

**REPLY:** We thank the reviewer for his/her comment. This is precisely what we do in our study, and the 14 August 2013 event is selected as it features dusty and convective conditions in the country on a day for which observational data is available for model evaluation, as noted in lines 394-397. In this work, and

through sensitivity experiments with the Weather Research and Forecasting (WRF) model, we explore the changes in atmospheric circulation, convection, radiation and precipitation to different aerosol loadings and properties, considering both an idealized and scaled version of a climatological distribution of aerosols. Examples of this are listed below:

- *Circulation & Convection*: Figs. 6a-b for ERA-5 and the WRF-3 simulation; Fig. A1 for runs WRF-4 to 5; Figs. 9c-d for simulations WRF-5 and 9; and Figs. 10c-d for runs WRF-6 to 8;

- *Precipitation & Convection*: Fig. 8 and S2 for all WRF simulations;

- *Radiation*: Figs. 9a-b for simulations WRF-5, 6 and 9; Fig. 10a-b for simulations WRF-6 to 8.

Besides, we compare the WRF predictions with in-situ measurements at the location of 35 weather stations spread out over the UAE (Fig. 1c). We present the results for the diurnal cycle for runs WRF-1 to WRF-4 in Fig. 7, and for all simulations we give the skill scores for the full day in Table S1.

However, and as stressed by the reviewer's previous comments, we agree that in the previous version of the manuscript we have not clearly highlighted our findings in the text, which is a cause for confusion. In the revised version of the paper we have done so in the Discussion and Conclusions section (lines 790-853) and in the Abstract (lines 20-34).

3.  In WRF simulations of this manuscript, how to consider the potential influences of environmental field (e.g. wind speed field, air humidity field), and vertical convection on the ARI, ACI, circulation, convection activity, and precipitation, etc?

**REPLY:** We thank the reviewer for his/her comment. In the Weather Research and Forecasting (WRF) model simulations conducted here, the aerosol-radiation interaction (ARI) and aerosol-cloud interaction (ACI) effects directly impact the environmental fields and vice-versa: i.e., they modulate the meteorological conditions where aerosols / clouds are present, and the modified atmospheric state influences the ARI and ACI. If the goal is to isolate the one-way interaction between the meteorological fields and the ARI/ACI effects, another modelling approach would have to be considered, such as the piggybacking framework (e.g. Grabowski, 2019). Such an analysis is beyond the scope of this study. We have stated this in the text (lines 871-875). As we highlight in the Introduction (lines 156-159), the goals of this work are twofold: (i) investigate the added value of incorporating aerosols on a dusty convective summertime event in a hyperarid region and account for their interactions with convection, and (ii) explore the sensitivity of the WRF model's response to changes in aerosol loading and properties. We believe this is achieved through the sensitivity experiments conducted in our study, with the results summarized in lines 790-835.

4.  The English written of this whole manuscript need to be greatly improved.

**REPLY:** We agree with the reviewer. In the revised version of the article we have rephrased poorly written and potentially confusing sentences, such as those in lines 24-27, 424-430, 579-583, 655-657, 752-754 and 787-788.

**Minor comments:**

1. **Page 3, lines 58-60:** 'Dust has been shown to have an important impact on the climate system, in particular on the atmosphere (e.g. Min et al., 2014; Liu et al., 2019; Francis et al., 2020), ocean (e.g. Evan et al., 2012) and cryosphere (e.g. Francis et al., 2018) dynamics.'

⇒ Please delete all the 'e.g.' in the cited literatures, and modify the other places in the context.

 **REPLY:** We thank the reviewer for his/her comment and have updated the text accordingly.

2.  When talking about the direct and semi-direct radiative effects of aerosols, the authors could cite other references.

[1] Li Z., Y. Wang, J. Guo, et al. 2019: East Asian study of tropospheric aerosols and their impact on regional clouds, precipitation, and climate (EAST-AIR(CPC)). Journal of Geophysical Research: Atmospheres. 124 (23), 13026-13054. DOI: 10.1029/2019JD030758.

[2] Wang W., J. Huang, P. Minnis, et al. 2010: Dusty cloud properties and radiative forcing over dust source and downwind regions derived from A-Train data during the Pacific Dust Experiment. Journal of Geophysical Research: Atmospheres. 115 . DOI:10.1029/2010JD014109.

**REPLY:** We thank the reviewer for his/her suggestion. We now cite the referred studies in the text (line 45).

**REFERENCES:**

Branch, P., Behrendt, A., Gong, Z., Schwitalla, T. and Wulfmeyer, V. (2020) Convection Initiation over the Eastern Arabian Peninsula. Meteorologische Zeitschrift, 29, 67-77. https://doi.org/10.1127/metz/2019/0997.

Fonseca, R., Francis, D., Nelli, N. and Thota, M. (2021) Climatology of the heat low and intertropical discontinuity in the Arabian Peninsula. International Journal of Climatology, 1-26. https://doi.org/10.1002/joc.7291.

Francis, D, Temimi, M, Fonseca, R, et al. On the analysis of a summertime convective event in a hyperarid environment. Q J R Meteorol Soc. 2021; 147: 501– 525. https://doi.org/10.1002/qj.3930

Grabowski, W. W. (2019) Separating physics impacts from natural variability using piggybacking technique. Advances in Geosciences, 49, 105-111. https://doi.org/10.5194/adgeo-49-105-2019.

Liu, L., Cheng, Y., Wang, S., Wei, C., Pohlker, M. L., Pohlker, C., Artaxo, P., Shrivastava, M., Andreae, M. O., Poschl, U. and Su, H. (2020) Impact of biomass burning aerosols on radiation, clouds, and precipitation over the Amazon: relative importance of aerosol–cloud and aerosol–radiation interactions. Atmospheric Chemistry and Physics, 20, 13283-13301. https://doi.org/10.5194/acp-20-13283-2020.

Roshan, D. R., Koc, M., Isaifan, R., Shahid, M. Z. and Fountoukis, C.: Aerosol Optical Thickness over Large Urban Environments of the Arabian Peninsula – Speciation, Variability, and Distributions. Atmosphere, 10(5), 228. https://doi.org/10.3390/atmos10050228, 2019.

Steinhoff, D. F., Bruintjes, R., Hacker, J., Keller, T., Williams, C., Jensen, T., Al Mandous, A. and Al Yazeedi, O. A. (2018) Influences of the Monsoon Trough and Arabian Heat Low on Summer Rainfall over the United Arab Emirates. Monthly Weather Review, 146(5), 1383-1403. https://doi.org/10.1175/MWR-D-17-0296.1.

Ukhov, A., Mostamandi, S., da Silva, A., Flemming, J., Alshehri, Y., Schevchenko, I. and Stenchikov, G.: Assessment of natural and anthropogenic aerosol air pollution in the Middle East using MERRA-2, CAMS data assimilation products, and high-resolution WRF-Chem model simulations. Atmospheric Chemistry and Physics, 20, 9281-9310. https://doi.org/10.5194/acp-20-9281-2020, 2020.

---

## Author Comment (AC2)

**ACP-2021-597: Convection-Aerosol Interactions in the United Arab Emirates: A Sensitivity Study**

**By Fonseca et al. (2021)**

**Reply to reviewer #2's comments:**

This manuscript attempts to use the WRF model to assess the convection-aerosol interactions over the UAE in summer. Unfortunately, there are some major flaws with the manuscript in its present form, and I have to recommend that the manuscript be rejected.

**REPLY:** The authors would like to thank the reviewer for his/her valuable comments/suggestions, which helped to improve the quality of the manuscript. Below we address the reviewer's queries one by one, both the major flaws and the specific comments, highlighting in the text where changes were made.

There are three critical issues with the paper.

**1.** The first is a lack of clear scientific question and focus. The title states that the topic of the paper is convection-aerosol interaction, but which part of that interaction is the key scientific question here? The introduction very broadly touched upon the impacts of ARI and ACI on the lifetimes and precipitations of MCS, but no key scientific question is brought forth. The abstract even mentions the impacts of nudging in the outer model domains, which further confuses the reader. Also, there are way too many sensitivity experiments. What is the purpose of testing (very) different aerosol composition assumptions in the model, if the point was to assess ACI in a particular case?

**REPLY:** We thank the reviewer for his/her comment. First and foremost, we have modified the title of the paper to reflect its main objective: investigate the sensitivity of the model-predicted convection to the aerosol loading and aerosol properties/composition. Following the reviewer's inputs, we have shortened it by (i) relegating the nudging discussion which we agree is not an essential part of the manuscript to an Appendix (lines 258-288) and therefore reduced the number of sensitivity experiments, and (ii) simplifying the discussion of the results focusing on the main findings (lines 790-853). We have also added a paragraph to the Discussion and Conclusions section on our recommendations for modelers targeting summertime convective events in this region, a very important take-home message (lines 837-853). We believe the manuscript's readability has improved and it has a clear focus now, with the discussion on the sensitivity to the aerosol composition fitting within the study's scope.

**2.** Secondly, the methodology used in this study is not appropriate for the question it appears to want to address. If the purpose is to investigate the 'interaction of convection and aerosol', then in the model, aerosols and cloud microphysics and dynamics should be allowed to 'interact' in a physically-realistic or reasonable way. Instead, the authors used a WRF model and implemented an 'aerosol-aware' cloud microphysics, which really does not allow aerosol and convection to 'interact' with each other. The use of assumed aerosol loading as initial condition and then allow them to be advected in no way physically represent the locations and strengths of the aerosols relative to the convective systems, as evidenced in Figs

2 and 5. Many of the assumptions (e.g., 30% dust in the radiation calculation; conversion of Ns to PM10; etc) are simply wrong.

**REPLY:** We thank the reviewer for raising this issue and apologize for not having explained in the text how the Thompson-Eidhammer aerosol-aware scheme works. The equations below which describe the temporal evolution of the number concentration of "water friendly" ($N_{wfa}$) and "ice friendly" ($N_{ifa}$) aerosols, and are taken from Thompson and Eidhammer (2014), their equations (3) and (4), respectively.

$$\frac{dN_{wfa}}{dt} = -\binom{rain, snow, graupel}{collecting\ aerosols} - \binom{homogenous\ nucleated}{deliquesced\ aerosols} - (CCN\ activation)$$

$$+\binom{cloud\ and\ rain}{evaporation} + \binom{surface}{emissions} \quad (1)$$

$$\frac{dN_{ifa}}{dt} = -\binom{rain, snow, graupel}{collecting\ aerosols} - (IN\ activation) + \binom{cloud\ ice}{sublimation} + \binom{surface}{emissions} \quad (2)$$

The most relevant aspects of the scheme discussed in Thompson and Eidhammer (2014) and, for "ice friendly" aerosols also in Su and Fung (2018), are summarized below:

➤ The nucleation of cloud droplets from $N_{wfa}$ is done through a lookup table with the activation fraction a function of parameters such as the WRF-predicted temperature, updraft speed, number of available aerosols, and predefined values of the hygroscopicity parameter and the aerosol's mean radius;

➤ Once nucleated, the aerosols are removed from $N_{wfa}$, third term on the right-hand-side (RHS) of equation (1), but can be restored via hydrometeor evaporation, fourth term in equation (1). Aerosols can also be removed by precipitation scavenging, first term in equations (1) and (2);

➤ For "water friendly" aerosols, and when a climatological-based distribution is employed, a constant surface emission forcing is added in the lowest model layer based on the starting near-surface aerosol concentration. A similar contribution is not considered for the "ice friendly" aerosols in the present version of the scheme, i.e. the last term on the RHS of equation (2) is set to zero;

➤ The nucleation of dust particles into ice crystals occurs in the presence of supersaturation with respect to ice. Depending on the relative humidity with respect to water, condensation, immersion freezing (i.e. ice nucleation by particles immersed in supercooled water) and deposition nucleation (i.e. formation of ice from supersaturated water vapour on an insoluble particle without the prior formation of liquid) can occur. These processes are accounted for by the second term on the RHS of equation (2);

➤ The freezing of homogeneous nucleated deliquesced hygroscopic aerosols is accounted for, with the decrease in $N_{wfa}$ represented by the second term on the RHS of equation (1), while the freezing of existing water droplets is more effective in the presence of higher amounts of dust aerosols. Cloud ice sublimation returns the aerosols to $N_{ifa}$, third term on the RHS of equation (2).

As the reviewer can see, aerosols and convection *do* interact with each other in the model, as besides transport we account for their emission and deposition. We have now made this clear in the text (lines 258-

288). While we agree that some approximations and simplifications have been made in the scheme, this "aerosol-aware" version of the Thompson cloud microphysics scheme has been widely used for convection-aerosol interaction studies such as in Alizadeh-Choobari and Gharaylou (2017). It has the advantage of being computationally cheaper than more sophisticated models like the WRF model with chemistry (WRF-Chem; Grell et al., 2005), as noted in the text (lines 221-224). This makes it possible to implement it in operational forecasts which is particularly pertinent for regions such as the UAE located next to major aerosol sources. We thank the reviewer again for his/her comment on this.

**3.** Thirdly, because the manuscript is lacking focus, it is also extremely long, without apparent need to be that way. For example, the verification diagnostics presented in section 2.4 are fairly standard; there is probably no need to elaborate. The discussion on the effects of nudging and the effects of assuming much of the aerosols to be carbonaceous is very confusing and not related to the topic at hand.

**REPLY:** We thank the reviewer for his/her comment, and fully agree that the manuscript is very long and needs to be shortened with the message sharpened. To that end, and following his/her suggestion, we have relegated the discussion of the nudging effects to an Appendix (a total of nine WRF simulations are now considered), and have added to the Discussion and Conclusions section a paragraph on the take-home messages to potential future readers of this study (lines 837-853). We believe the simulations on the sensitivity to the aerosol properties should remain in the text, as they are an integral part of this study (the focus is both on the aerosol loading and composition as now clearly mentioned in the text/title) as noted in the reply to the reviewer's critical comment #1. Likewise, some of the verification diagnostics employed are not standard (e.g. the normalized error variance, $\alpha$, and the variance similarity, $\eta$), so they have to be defined in the text. We do so first mathematically and then with a sentence to explain what information the skill score conveys and the optimal value. In any case, we have relegated Table 3 as supplementary material. We have also simplified the discussion of the results to make the manuscript easier to follow. Examples of this can be seen in lines 504-518, 668-683 and 790-835. We would like to thank the reviewer again for raising this issue.

Specific comments:

**1.** Lines 12-13: "Both an idealised and ... are considered": This sentence is unclear. Please revise.

**REPLY:** We thank the reviewer for raising the issue and have rephrased the phrase accordingly (lines 13-14).

**2.** Lines 24-28: "In particular, ... 51 W/m2.": This sentence is extremely long and unclear. Please revise.

**REPLY:** We agree with the reviewer that this sentence is indeed very long. In the revised version of the manuscript we broke it into two and rephrased it for clarity (lines 20-24).

**3.** Line 28: Not sure what "the former" and "the latter" refer to.

**REPLY:** We are referring to the simulations where the aerosol composition and aerosol loading are changed, respectively. We have rephrased the sentence in the revised version of the paper (lines 24-25).

**4.** Lines 51-52: The increased number of smaller cloud droplets mostly lead to more scattering (hence higher albedo and optical depth). This is not the same as 'reducing the radiative window'. Also, this statement is missing reference.

**REPLY:** We have rephrased the sentence accordingly and added a reference (lines 49-51).

**5.** Line 54: Albrecht effect: missing reference.

**REPLY:** We have added a reference for the Albrecht effect (line 53).

**6.** Line 248: "Nwfa and Nifa ... evolve during the course of the model integration": How is this achieved? Do you simulate the emission/transport/deposition of the Ns? Or do you prescribe how N changes with time? If the latter, how do you ensure that this correctly represents the response of N to meteorology. More importantly, how do you know that you are not forcing the cloud microphysics to do things that you wish to see?

**REPLY:** Following the reply to the reviewer's critical comment #2, we now make it abundantly clear in the text how the "aerosol-aware" version of the Thompson scheme works (lines 258-288). Yes, the model simulates the emission and deposition of the aerosol number concentration besides its transport, all done in a physically-sound way.

**7.** Lines 249-250: "...dataset on a monthly time-scale ... downloaded from the model's website": What is this dataset based on? Is it based on a model calculation or some kind of satellite data inversion?

**REPLY:** We thank the reviewer for raising this issue and apologize for the lack of clarity in the text. This is the 7-year (2001-2007) climatological dataset we refer to in lines 230-232, which is obtained from a global model simulation in which aerosols are emitted by natural and anthropogenic sources are modelled using a bin scheme. This dataset is therefore a model product but the predicted aerosol optical depth and Angstrom exponent compare well against satellite estimates in particular in this region, as discussed in Colarco et al. (2010). We have now stated this in the text (lines 249-254).

**8.** Line 252: So Nwfa and Nifa are first set with initial conditions and then allowed to advect and diffuse? How do you ensure that the transported Ns are realistic? Also, the scavenging of Ns (both hygroscopic and non-hygroscopic) by precipitation are not considered?

**REPLY:** As stated in the reply to the reviewer's critical comment #2, and besides the transport, the emission and deposition of aerosols are accounted for in the model, including the scavenging due to precipitation. This is now clearly stated in the text (lines 258-288).

**9.** Lines 249-255: The description of the aerosol settings in the sensitivity experiments is unclear. I have a hard time following what assumptions are made. Please consider revising.

**REPLY:** We thank the reviewer for raising this issue. In the referred sentences we are mixing the idealized set up with the climatological initialization for aerosols which is indeed confusing. We have revised the text for clarity (lines 234-256).

**10.** Line 257: Does the number of non-hygroscopic aerosol affect the number of ice nuclei? This is not a default option in WRF. What parameterization is used to describe this sensitivity?

**REPLY:** Yes, it does, as noted in the reply to the reviewer's critical comment #2 and given by the second term on the RHS of equation (3). This option is only available in the Thompson-Eidhammer "aerosol-aware" cloud microphysics scheme which we employ in the simulations discussed in this study. We now describe the scheme in more detail in the text for clarity (lines 279-283).

**11.** Lines 264-265: "...assumes a mixture of 70% water soluble and 30% dust-like aerosols": Is this a reasonable assumption for this case, where most of the aerosols were dust? More importantly, is this assumption consistent with the prescribed hygroscopic/non-hygroscopic aerosol numbers?

**REPLY:** We thank the reviewer for raising this issue. We would like to stress that in this manuscript we aim at investigating the sensitivity of the model-predicted convection and atmospheric fields to the aerosol loading and composition, so as to issue recommendations to WRF modelers as to which set up may work best for a given simulation. Our goal is *not* to mimic the observed aerosol conditions, as we do not have relevant measurements that will allow us to do so.

In the comment above, the reviewer is referring to the assumptions made regarding the aerosol properties, in particular the single-scattering albedo, asymmetry factor and Angstrom exponent. Here we simply tested the three options available by default in WRF, denoted as aerosol models in the text, with one being that referred by the reviewer. A customization of the aerosol properties for our event is not possible for the aforementioned reasons. Regarding the reviewer's second question, we agree that the fraction of hygroscopic and non-hygroscopic aerosols is in closer agreement with the assumptions made in some of the aerosol models but not so much in others. In any case, we would like to note that, at a given grid-point, the percentage of each varies during the simulation, as seen in Figs. 5a-b, meaning that no one aerosol model works well all the time. Hence, we believe it is acceptable to test the different options for the event targeted in this study. We hope the reviewer agrees with us on this. Nevertheless, we have noted the caveats in the text (lines 307-311).

**12.** Lines 278-281: Is the effect of nudging in the outer domains one of the scientific questions you want to address in this study? If not, and if nuding is necessary to capture the large-scale atmospheric dynamics, then it should be included in all the key experiments. Otherwise there are simply too many experiments without a clear, key scientific question.

**REPLY:** We thank the reviewer for raising this issue. As stated in the reply to the reviewer's critical comment #3, we agree that the paper is too long and lacks a clear focus. Following his/her comment, we have relegated the nudging experiments to Appendix A (lines 1324-1379), and focused on the sensitivity to the aerosol loading and properties in the main text, the central goal of the study. We believe the paper is more targeted now and easier to follow.

**13.** Lines 312-314: Why use MERRA-2 aerosol product for this study? Has the AOD in MERRA-2 been evaluated over this part of the world, particular since the surface is bright?

**REPLY:** Yes, MERRA-2 has been found to perform well in the Middle East when compared to satellite-derived and ground-based measurements as concluded e.g. by Roshan et al. (2019) and Ukhov et al. (2020). We have also compared the 3-hourly MERRA-2 PM10 with that observed hourly at air quality stations and found a good agreement between the two, despite the still coarse spatial ($0.625° \times 0.5°$) resolution of the reanalysis dataset and inherent model biases. The reviewer can see this in Fig. R1 below for two stations, one in the northeastern (Burairat) and another in western (Gayathi School) UAE, on 14 August 2013.

(a)                                                          (b)

[Figure]

**Figure R1**: Observed (red line) and MERRA-2 (blue dots) PM10 concentration ($\mu g\,m^{-3}$) for the (a) Burairat station in northeastern UAE and (b) Gayathi School station in western UAE on 14 August 2013.

As AOD estimates from ground and satellite assets exhibit gaps and missing data on this day due to the extensive cloud cover (Figs. 2a-c), we decided to use the referred reanalysis dataset that explicitly accounts for aerosols and their interactions with the climate system and is found to perform well in the region. We have now stated this in the text (lines 347-349 and 504-511).

**14.** Lines 351-358: In fig2a-c, it appears that the dusts are mostly in the Northeastern part of the domain, but the MERRA-2 AODs are mostly in the central/southern part of the domain. Which one is more accurate and how do you reconcile the discrepancy? More importantly, which one is more consistent with the assumed Ns in the simulation?

**REPLY:** We would like to note that in the RGB images, Figs. 2a-c, and because of the extensive cloud cover, one cannot see the dust over the UAE, which can be emitted in association with cold pools as is common in desert regions (e.g. Bou Karam et al., 2014). Therefore, a direct comparison with MERRA-2 (and WRF) is not possible. While the AOD in MERRA-2 is lower over Iran when compared to the UAE, it is non-zero in the regions shaded in pink in the RGB images, so the two products are not necessarily inconsistent. A comparison of Figs. 5a-b with Figs. 2g-i shows that WRF also predicts lower amounts of dust over Iran as MERRA-2, while over the UAE it underestimates the dust loading compared to the reanalysis dataset, as seen in Fig. 6d. This is discussed in the text (lines 504-518).

**15.** Fig 2a-c: How are these figures colored? If this is indeed 'RGB' (i.e., real color) images, why would the dust be pink and the clouds orange/brown/black? Clearly, some other type of of processing has been applied.

**REPLY:** We thank the reviewer for raising this issue. Indeed, these are not raw RGB images but processed (false-colour) images to highlight specific features such as dust, sand and clouds in contrasting colours, following Banks et al. (2019). False-colour RGB images such as these are widely used in aerosol-related studies such as Francis et al. (2020). We have stated this in the text (lines 338-339).

**16.** Section 4.1: The aerosol loading simulated here are neither consistent with the RGB plots or the MERRA-2 AOD shown in Fig 2.

**REPLY:** As highlighted in the reply to reviewer's specific comment #14, we cannot directly compare the RGB images with the WRF and MERRA-2 plots. Besides, the dust over Iran in the RGB plots is not necessarily inconsistent with the MERRA-2 AODs in Fig. 2 and the WRF plots in Figs. 5a-b. We do accept that there are inconsistencies between MERRA-2 and WRF, and discussed them in the text in section 4.1 (lines 504-518).

**17.** Figure 5: The labeling is extremely confusing. Some subplots are not labeled, and which ones are (c) and (d)? The caption says: "The aerosol concentration in panels (c) and (d) is scaled as in panels (a) and (b)", which I do not understand.

**REPLY:** We apologize for the confusing caption of Figure 5 and have rephrased it in the revised version of the manuscript. Panels (a) and (b) have six subplots each and panels (c) and (d) have two subplots each which have a title but not a label so as to make the interpretation of the figure easier. Nevertheless, we have improved the readability of the figure in the updated version of the paper.

**18.** Section 4.1: The conversion of WRF 'simulated' aerosol loading to PM10 is inappropriate. The Ns here are not realistic aerosol loading. They are first prescribed as initial condition and then allowed to be transported. In no way were the Ns physically related to dust in the model.

**REPLY:** As noted in the reply to the reviewer's critical comment #2, we believe the aerosol concentration used here is realistic, as besides advection we also account for emission and deposition. In any case, we agree that the comparison between the WRF "ice friendly" aerosols and the observed PM10 at the location of the 11 weather stations is not correct as several assumptions have been made, some of which are questionable such as (i) considering that PM10 is just due to the dust in the model, (ii) comparing the WRF model-layer concentration with that observed at a single height/point for pollutants that exhibit a high temporal and spatial variability, and (iii) taking a fixed radius and density for the PM10 particles. We have taken out this comparison from the article, leaving just the evaluation of the WRF-predicted AOD with that given by MERRA-2 (lines 511-518), a more straightforward comparison.

**REFERENCES:**

Alizadeh-Choobari, O. and Gharaylou, M.: Aerosol impacts on radiative and microphysical properties of clouds and precipitation formation. Atmospheric Research, 185, 53-64. https://doi.org/10.1016/j.atmosres.2016.10.021, 2017.

Banks, J. R., Hunerbein, A., Heinold, B., Brindley, H. E., Deneke, H. and Schepanski, K.: The sensitivity of the colour of dust in MSG-SEVIRI Desert Dust infrared composite imagery to surface and atmospheric conditions. Atmospheric Chemistry and Physics, 19, 6893-6911. https://doi.org/10.5194/acp-19-6893-2019, 2019.

Bou Karam, D., Williams, E., Janiga, M., Flamant, C., McGraw-Herdeg, M., Cuesta, J., Auby, A. and Thorncroft, C.: Synoptic scale dust emissions over the Sahara desert initiated by a moist convective cold pool in early august 2006. Quarterly Journal of the Royal Meteorological Society, 140(685), 2591-2607. https://doi.org/10.1002/qj.2326, 2014.

Colarco, P., da Silva, A., Chin, M. and Diehl, T.: Online simulations of global aerosol distributions in the NASA GEOS-4 model and comparisons to satellite and ground-based aerosol optical depth. J. Geophys. Res., 115, D14207. https://doi.org/10.1029/2009JD012820, 2010.

Francis, D., Fonseca, R., Nelli, N., Cuesta, J., Weston, M., Evan, A. and Temimi, M.: The atmospheric drivers of the major Saharan dust storm in June 2020. Geophysical Research Letters, 47, e2020GL090102. https://doi.org/10.1029/2020GL090102, 2020.

Grell, G. A., Peckham, S. E., Schmitz, R., McKeen, S. A., Frost, G., Skamarock, W. C. and Eder, B.: Fully coupled "online" chemistry with the WRF model. Atmos. Environ., 39, 6957-6975. https://doi.org/10.1016/j.atmosenv.2005.04.027, 2005.

Roshan, D. R., Koc, M., Isaifan, R., Shahid, M. Z. and Fountoukis, C.: Aerosol Optical Thickness over Large Urban Environments of the Arabian Peninsula – Speciation, Variability, and Distributions. Atmosphere, 10(5), 228. https://doi.org/10.3390/atmos10050228, 2019.

Su, L. and Fung, J. C. H.: Investigating the role of dust in ice nucleation within clouds and further effects on the regional weather system over East Asia – Part 1: model development and validation. Atmospheric Chemistry and Physics, 18, 8705-8725. https://doi.org/10.5194/acp-18-8707-2018, 2018.

Thompson, G. and Eidhammer, T.: A study of aerosol impacts on clouds and precipitation development in a large winter cyclone. J. Atm. Sci., 71, 3636-3658. https://doi.org/10.1175/JAS-D-13-0305.1, 2014.

Ukhov, A., Mostamandi, S., da Silva, A., Flemming, J., Alshehri, Y., Schevchenko, I. and Stenchikov, G.: Assessment of natural and anthropogenic aerosol air pollution in the Middle East using MERRA-2, CAMS data assimilation products, and high-resolution WRF-Chem model simulations. Atmospheric Chemistry and Physics, 20, 9281-9310. https://doi.org/10.5194/acp-20-9281-2020, 2020.